# Tradeoff breaking as a model of evolutionary transitions in individuality and limits of the fitness-decoupling metaphor

Pierrick Bourrat[1,2]*[†], Guilhem Doulcier[1,3]*[†], Caroline J Rose[4], Paul B Rainey[5,6], Katrin Hammerschmidt[7]*

[1]Philosophy Department, Macquarie University, Sydney, Australia; [2]Department of Philosophy & Charles Perkins Centre, The University of Sydney, Sydney, Australia; [3]Department of Evolutionary Theory, Max Planck Institute for Evolutionary Biology, Plön, Germany; [4]Centre d'Écologie Fonctionnelle et Évolutive (CEFE), CNRS, Montpellier, France; [5]Laboratoire Biophysique et Évolution, CBI, ESPCI Paris, Université PSL, CNRS 75005 Paris, France, Paris, France; [6]Department of Microbial Population Biology, Max Planck Institute for Evolutionary Biology, Plön, Germany; [7]Institute of Microbiology, Kiel University, Kiel, Germany

**\*For correspondence:**
p.bourrat@gmail.com (PB);
guilhem.doulcier@normalesup.org (GD);
katrinhammerschmidt@googlemail.com (KH)

[†]These authors contributed equally to this work

**Abstract** Evolutionary transitions in individuality (ETIs) involve the formation of Darwinian collectives from Darwinian particles. The transition from cells to multicellular life is a prime example. During an ETI, collectives become units of selection in their own right. However, the underlying processes are poorly understood. One observation used to identify the completion of an ETI is an increase in collective-level performance accompanied by a decrease in particle-level performance, for example measured by growth rate. This seemingly counterintuitive dynamic has been referred to as fitness decoupling and has been used to interpret both models and experimental data. Extending and unifying results from the literature, we show that fitness of particles and collectives can never decouple because calculations of fitness performed over appropriate and equivalent time intervals are necessarily the same provided the population reaches a stable collective size distribution. By way of solution, we draw attention to the value of mechanistic approaches that emphasise traits, and tradeoffs among traits, as opposed to fitness. This trait-based approach is sufficient to capture dynamics that underpin evolutionary transitions. In addition, drawing upon both experimental and theoretical studies, we show that while early stages of transitions might often involve tradeoffs among particle traits, later—and critical—stages are likely to involve the rupture of such tradeoffs. Thus, when observed in the context of ETIs, tradeoff-breaking events stand as a useful marker of these transitions.

## Editor's evaluation

This article makes two important, independent contributions to the multicellularity/major transitions literature. First, it sheds light on the concept of 'fitness decoupling', providing a strong mathematical foundation for the claim that the fitnesses of cells and groups cannot be formally decoupled during an evolutionary transition in individuality. Second, the article proposes using the evolution of tradeoff-breaking traits as an indication that an evolutionary transition has occurred. This dovetails well with existing fitness-based approaches, extending the toolkit of those studying major evolutionary transitions.

## Introduction

Evolutionary transitions in individuality (ETIs) are events of major significance in the history of life. They begin with lower-level entities (particles) and complete when higher-level entities (collectives) acquire properties sufficient to participate directly in the process of evolution by natural selection (see Appendix 1, 'Glossary' for definitions of technical terms). ETIs of particular note include the evolution of chromosomes (from genes), the eukaryotic cell (from an ancestral eubacterium and archaebacterium), and multicellularity (from single cells) (*Buss, 1987*; *Jablonka, 1994*; *Maynard Smith and Szathmáry, 1995*; *Michod, 1999a*; *Bourke, 2011*; *Calcott and Sterelny, 2011*; *Bouchard and Huneman, 2013*; *West et al., 2015*; *van Gestel and Tarnita, 2017*). Here, we focus attention on the transition from cells to multicellular life.

Evolutionary dynamics underpinning the transition to multicellularity have proven difficult to capture; however, influential for theory—and guiding experimental analysis—has been the concept of 'fitness decoupling', which posits that the fitness of a collective in the early stage of a transition is directly proportional to the fitness of its particles and that, as the evolutionary transition proceeds, collective fitness 'becomes decoupled from the fitness of its lower-level components' (*Michod and Nedelcu, 2003*, p. 96). This notion is based on the fact that the evolution of cooperation, division of labour, or conflict-mediating mechanisms seems to improve collective-level fitness at the cost of particle-level fitness. This phenomenon has been interpreted through the lens of an export-of-fitness framework, in which the evolution of such mechanisms marks 'transfer of fitness' from particle to collective levels (*Michod and Roze, 1999b*; *Michod and Nedelcu, 2003*; *Michod, 2005*; *Okasha, 2006*; *Michod et al., 2006b*; *Okasha, 2009*; *Folse and Roughgarden, 2010*; *Michod et al., 2003*; *Shelton and Michod, 2020*; *Michod, 2022*; *Doulcier et al., 2022*).

In recognising challenges associated with assigning fitness to particles, as opposed to collectives, and building from earlier work (e.g., *Michod, 2005*; *Michod, 2006a*), *Shelton and Michod, 2014* proposed the idea of 'counterfactual fitness'. They defined the counterfactual fitness of a collective as the fitness that this collective would have if there were no collective-dependent effects on the interactions of particles within the collective. This approach can be regarded as a way of apportioning fitness components to different levels: the counterfactual component corresponds to the particle level, while the collective-dependent effects correspond to the collective level. Counterfactual fitness was later integrated in the fitness decoupling, fitness transfer or export-of-fitness framework (explicitly in *Shelton and Michod, 2020*, but see *Michod, 2005*; *Michod, 2006a*, for the precursor of this idea). Accordingly, fitness decoupling is seen to have occurred when the collective-level-dependent component of fitness increases and the counterfactual component goes to zero. While useful, in that the idea of counterfactual fitness permits accounting for the functional integration or mutual interdependence of the particles within an entity, which is an important characteristic of an individual (see *Godfrey-Smith, 2009*), its use in the context of fitness decoupling leaves unresolved the mechanism by which fitness at one level can be exported to the other. A primary goal of this article is to add clarity to these discussions.

Although the idea of fitness transfer or decoupling, whether understood literally or counterfactually, seems to be intuitive, and empirical tests straightforward, it has been suggested that this might not be the case. The notion of fitness and its use in evolutionary thinking is generally problematic (as pointed out, among others, by *Doebeli et al., 2017*). In addition, actual particle and collective fitness are often 'mathematically equivalent ways of bookkeeping' (as pointed out by *Shelton and Michod, 2014*, p. 457, in the context of their model). Further, measurement of fitness in experiments—particularly in the context of ETIs, where fitness is usually compared between two organisational levels—is difficult (*Bourrat, 2015a*; *Bourrat, 2015b*; *Bourrat, 2021a*; *Bourrat, 2021b*). Experimental investigations of early stages in the transition to multicellularity (*Hammerschmidt et al., 2014*; *Rose et al., 2020*) are a good example thereof. Data from these experiments showed collective-level fitness to have improved while cell-level fitness declined over evolutionary time, giving the impression that the fitness of cells had decoupled from the fitness of collectives (see below for details). Although several proxies for fitness were obtained, such as maximum growth rate and competitive fitness (as is traditionally done in the field, see *Lenski et al., 1991*; *Wiser et al., 2013*; *Wiser et al., 2015*), for practical reasons, it was impossible to measure cell and collective fitness over the same timescale, which would have been required for a meaningful comparison (see Box 1). The notion of counterfactual fitness is also difficult to operationalise as no consensus method for its measure has emerged

## Box 1. Comparing fitnesses.

Determining whether the fitness of an entity is higher or lower than that of another entity requires obtaining commensurate fitness estimates for the two entities. In this box, we consider monomorphic lineages; thus, 'individual' or 'lineage' fitness can be used interchangeably.

A conventional fitness estimate is an entity's expected number of offspring after one generation. However, for this estimate to be adequate for the purpose of comparison, the generation time of the entities compared must be the same. If they are not, a different/same number of offspring per generation does not necessarily indicate a different/same fitness value. To see this point, suppose two entities $A$ and $B$, which double at each generation, such as two strains of bacteria. Everything else being equal, $A$ doubles at a *faster* rate than $B$ because it has a shorter generation time. Over the same absolute period, $A$ will have a higher number of descendants than $B$, and one could infer that $A$ is fitter than $B$. **Box 1—figures 1a and 1b** illustrate the situations, respectively, where a fitness comparison is made based on generational (and, thus, invalid) and absolute times (valid).

A more appropriate measure for the purpose of fitness comparison when generation times differ between the focal entities is the long-term growth rate of the population or the Malthusian parameter. These values are often computed from empirical actuarial tables using population projection models (see **Caswell, 1989**) or the repeated census of the population (e.g., in microbiology).

However, long-term growth rates can only be compared if they are made over the same set of events. Any observation where one entity appears fitter than the other, but for which a different set of events has been used (different timescale or different environment), could in reality be one where there is no difference between the two. In other words, it could be a spurious observation.

To take an example where the two environments, rather than the times over which reproductive outputs are measured, are different, consider two identical plants receiving different quantities of resources. The plant receiving more resources produces more seeds. Yet, this difference in reproductive output cannot lead to the conclusion that these two plants have different fitnesses where 'fitness' is associated with natural selection. In our example, because the two plants are initially identical, they necessarily have the same fitness. This situation is depicted in **Box 1—figure 1c and d**. In **Box 1—figure 1c**, the fitnesses are compared in two different environments—and, thus, the comparison is invalid—while in **Box 1—figure 1d**, the comparison is made in the same environment.

In some situations where one wants to make fitness comparisons, the environment presents fluctuations in time. In this case too, to be comparable, they must refer to the same set of events. For instance, if $A$ is in environment 1, and $B$ is successively in environments 1 and 2, the two resulting fitness values are not comparable because they do not inform one of the potential outcomes of competition in environment 1, in environment 2, or in a temporal succession of the two. This invalid comparison is represented in **Box 1—figure 1e**. Therefore, a condition for comparison, taking into account environmental change, is that the two organisms follow the same temporal succession of environments, as presented in **Box 1—figure 1f**. Note, crucially, that we assume here that whether an entity is in a given environment is independent of its type. If a dependence of the environment on the type exists, this environment effectively becomes an extended phenotype (**Lu and Bourrat, 2018**).

If the environmental changes are not deterministic, a weaker condition than the same temporal succession of environments is that the two organisms experience the same distribution of environments and transition probabilities between environments (steady-state) (see **Doulcier et al., 2021**). This type of scenario is not discussed in the main text (but see **Box 3**).

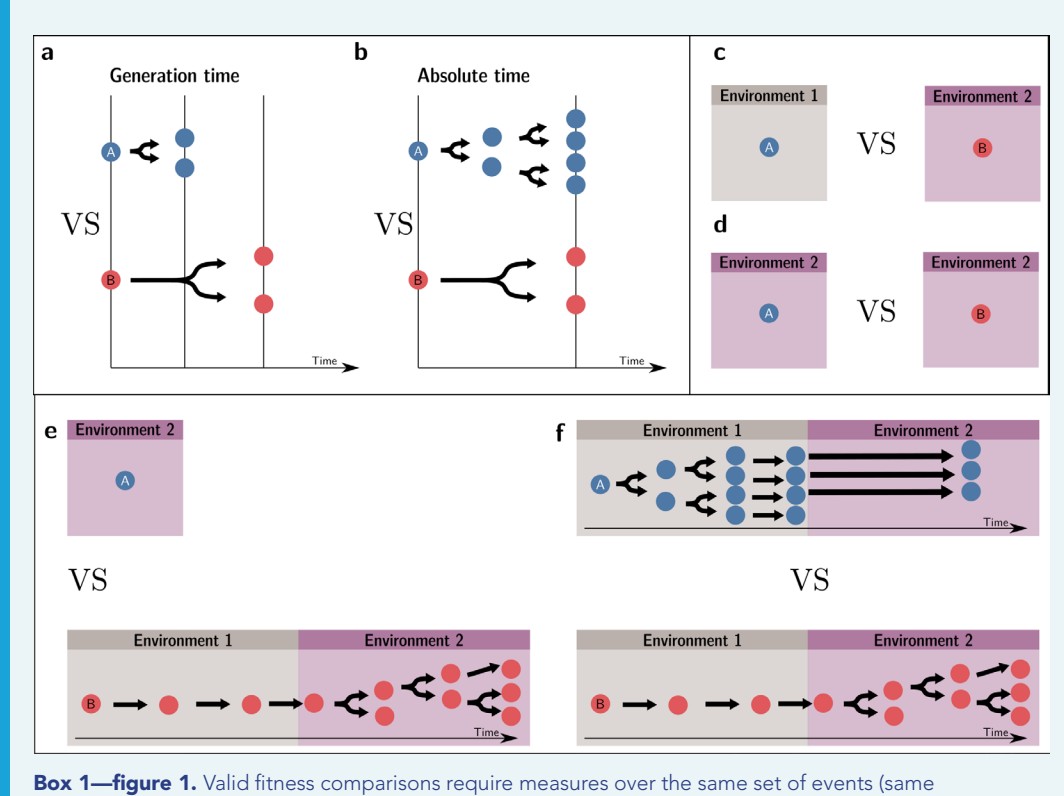

**Box 1—figure 1.** Valid fitness comparisons require measures over the same set of events (same environment and timescale).
Invalid comparisons: (**a, c, e**). Valid comparisons: (**b, d, f**).

from the literature. The relevance of various proxies that could be used, such as measuring fitness in a modified environment that prevents the formation of collectives (e.g., by shaking the culture medium) or measuring the fitness of genetically engineered mutants unable to form collectives (revertant for the collective-formation mutations), is yet to be assessed.

Building on this experimental approach and philosophical insights about valid fitness comparisons between levels of organisation from *Bourrat, 2015a* and *Bourrat, 2015b*, *Black et al., 2020* constructed a simple 'ecological scaffolding' model that showed how a minimal set of ecological conditions can produce evolutionary dynamics where particle and collective fitness appear 'decoupled'. Population structure creates a tradeoff between short-term growth through particle division and long-term growth through collective persistence. This led Black et al. to conclude that a 'fitness decoupling' observation can be explained in terms of a reduction of *short-term* particle growth rate coinciding with increased collective-level performance (over a *longer* timescale), rather than in terms of transfer of fitness between particles and collectives. If it is true that the fitness of cells and collectives cannot decouple, where does this leave our understanding of dynamic processes that underpin ETIs? The mechanistic approach with an ecological focus is one route (see *Black et al., 2020*; *Bourrat, 2022*; *Doulcier et al., 2020*), as is the ratcheting-mutation model proposed by *Libby et al., 2016*. However, there is also a need to identify hallmarks of transitions that might allow identification and experimental validation of ETIs.

We begin by elaborating on the notion of 'fitness decoupling' and related concepts such as 'export of fitness' and 'fitness transfer', showing how this idea has been used in both the theoretical and experimental literature (Section 1). In Section 2, we present a population projection model of an abstract proto-multicellular organism and show that particle and collective fitness are necessarily equal for all structured population dynamics models that reach a stable collective size distribution, generalising and unifying results from the literature (e.g., *Shelton and Michod, 2014*; *Bourrat, 2015a*; *Bourrat, 2015b*, *Bourrat, 2021a*; *Black et al., 2020*). In Section 3, we show that the intuited dynamics of transitions are captured best by the language of traits and tradeoffs among traits. Finally, we turn

to the challenge of defining essential features of ETIs and argue that events that break the tradeoff constraining the ancestral particle—hereafter referred to as 'tradeoff breaking'—stand as a marker of an ETI (Sections 4 and 5).

## Results
## 1. Challenges arising from the application of fitness-centred approaches to the study of ETIs

One classical way to characterise particle fitness is to measure long-term reproductive success under a given set of environmental conditions relative to other particles (*Pence and Ramsey, 2013*; *Doulcier et al., 2021*). In a more practical sense, fitness is often measured as a per capita growth rate—that is, the average number of offspring produced by an individual per unit of time (or per generation) (*Fisher, 1930*; *Metz et al., 1992*). Whenever a nested system (composed of particles assembled into collectives) is studied, it is possible to measure at least two kinds of 'fitnesses': the fitness of particles and that of collectives. To do so, population growth is tallied at each level.

Proponents of the export-of-fitness model and the concept of fitness decoupling argue that during an ETI, the fitness of particles and collectives of particles become 'decoupled' (*Michod and Roze, 1999b*; *Michod and Nedelcu, 2003*; *Michod, 2005*; *Shelton and Michod, 2009*; *Hanschen et al., 2015*; *Hanschen et al., 2017*; *Shelton and Michod, 2020*; *Davison and Michod, 2021*). More precisely, the two values are predicted not to change in the same way and even to change in different directions: collective-level fitness increases while particle-level fitness decreases (*Hanschen et al., 2017*). Note that they are not expected to necessarily become independent from one another. All that is required for them to become decoupled is that they are anticorrelated or even less correlated. In some stricter formulations of the concept, collective and particle fitnesses are said to be decoupled as soon as collective fitness is not proportional to the average fitness of its component particles (*Michod and Nedelcu, 2003*; *Okasha, 2006*). In this section, we review the theoretical and empirical arguments leading to this prediction.

The idea of fitness decoupling can be traced back to the study of the disruption of higher-level entities by the proliferation of lower-level entities composing them (see *Maynard Smith and Szathmáry, 1995*, pp. 7–8). If selection acts at two levels during an ETI, why would selection at the lower level (on molecules, cells, or organisms) not disrupt the effect of selection at the higher level (on chromosomes, multicellular organisms, or insect colonies)? Such 'conflicts' pose a challenge for integration at the higher level. This phenomenon is present at all organismal levels. For instance, cancerous cells proliferating at a higher rate than healthy cells pose a threat to organismal integrity (*Merlo et al., 2006*). Similarly, without suppression from the queen, egg-laying worker bees pose a threat to the integrity of the hive (*Amdam and Page, 2010*). Other examples include selfish genetic elements, which can sometimes produce harmful effects at the organism level (see *Werren, 2011*, for a review). As a consequence, one may prima facie expect evolutionary trajectories to be the result of opposing processes that would be both a hallmark and a significant hurdle for ETIs.

The idea of fitness decoupling was introduced by *Michod and Nedelcu, 2003* (pp. 66–67), to describe the ways conflicts between the higher and lower levels of organisation can be resolved during an ETI: "as the evolutionary transition proceeds, group fitness becomes decoupled from the fitness of its lower-level components […] This transfer and reorganization of fitness components from lower to higher-levels occurs through the evolution of cooperation and mediators of conflict that restrict the opportunity for within-group change and enhance the opportunity for between-group change." More generally, *Hanschen et al., 2017* note that "any trait that is costly at the lower level but beneficial at the group level enhances the fitness of the group at the expense of lower-level fitness and may therefore contribute to fitness decoupling and the emergence of indivisibility of the group". Okasha and Michod recast the notion of fitness decoupling in the multilevel selection 1/multilevel selection 2 (MLS1/MLS2) framework (*Michod, 2005*; *Michod and Nedelcu, 2003*; *Okasha, 2006*, p. 232). Okasha describes collective fitness during the three stages of an evolutionary transition (*Okasha, 2006*, p. 238): in the first stage, collective fitness is defined as average particle fitness (MLS1); second, collective fitness is defined as proportional to average particle fitness; finally, gradual decoupling occurs in the transition towards the third stage, where collective fitness is no longer proportional to particle fitness (MLS2). In the export-of-fitness framework of ETIs, collectives initially 'lack' individuality

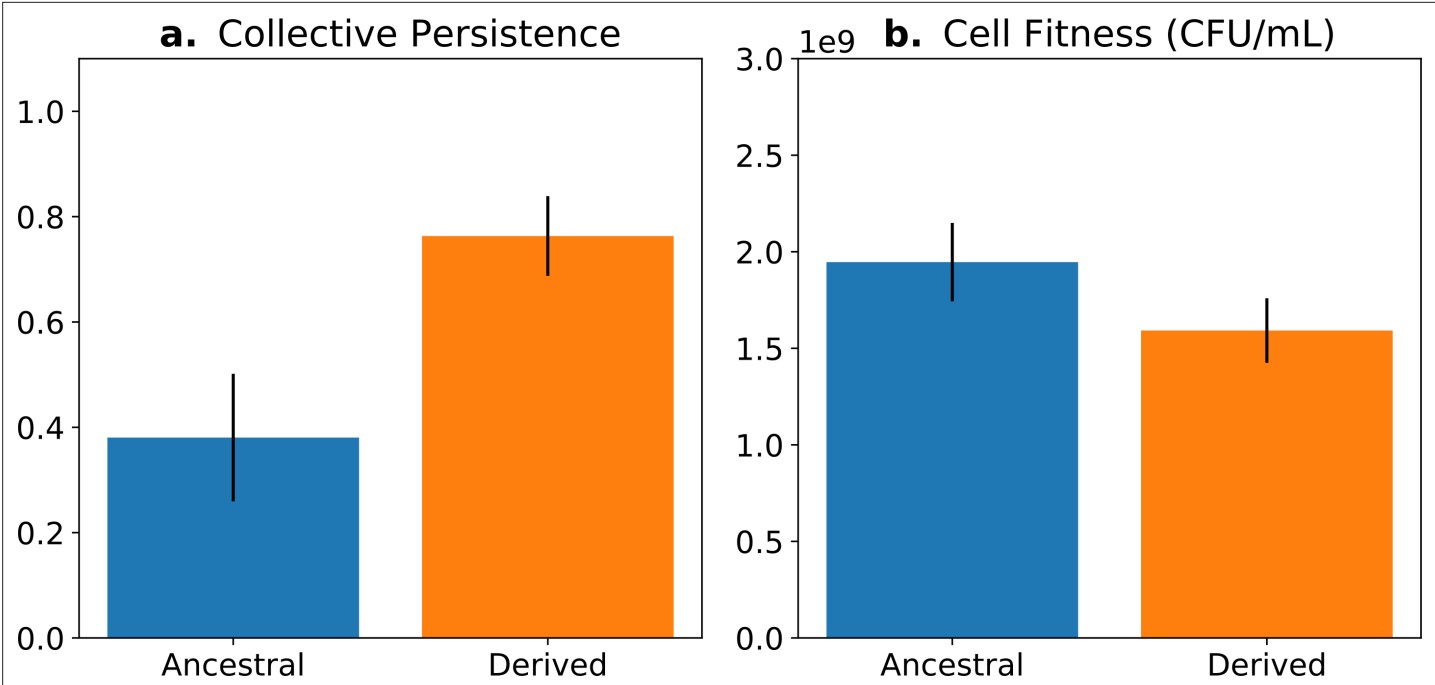

**Figure 1.** Fitness-decoupling observation in the *Pseudomonas* system. Comparison of collective-level persistence (measured as the proportion of collective persistence after one generation when competed against an ancestral reference strain) and cell (particle) fitness (measured as the number of cells comprising a collective) for ancestral (blue) and derived (orange) populations under a regime designed to promote an evolutionary transition in individuality. Error bars represent standard errors of the mean (based on n = 15 ancestral and n = 14 derived lineages, respectively). Redrawn from *Hammerschmidt et al., 2014* (Fig. 2), for ease of comparison with *Figure 5*. Protocol described and statistical analysis performed in *Hammerschmidt et al., 2014*, showing statistical significance between ancestral and derived collective persistence/cell fitness. Dataset published as *Rose et al., 2018*.

(fitness of particles and collectives are proportional) and 'gain' individuality once their fitness is 'transferred' from the underlying particles (*Michod and Roze, 1999b*; *Okasha, 2006*, pp. 234–235).

The concept of fitness decoupling has been regarded as one indicator for the ETI from cells to multicellular individuals (*Rainey and Kerr, 2010*; *Pichugin, 2015*; *Hanschen et al., 2015*; *Conlin et al., 2019*). In a study using experimental bacterial (*Pseudomonas fluorescens*) populations, *Hammerschmidt et al., 2014* and *Rose et al., 2020* propagated collectives over multiple generations, and then asked whether predicted increases in collective-level fitness were realised. This was achieved by competing derived collectives against ancestral collectives over the time frame of a single life cycle generation—the number of offspring collectives left over this period was greater in the derived populations (*Figure 1a*). Next, they sought understanding of cell-level fitness effects. Ideally, such assays would have been performed over an entire collective-level life cycle; however, for practical reasons, this is nigh impossible. Instead, various cell-level assays were conducted, including assessment of competitive ability in broth culture for the duration of part of the life cycle. Data from these experiments showed cell-level fitness to have declined in derived populations (*Figure 1b*), at least for the regime where lineages passaged through soma-like and germ-like phases, giving the impression that the fitness of cells had decoupled from the fitness of collectives (*Figure 1*).

Measures of cell and collective fitness must derive from analyses performed in the same reference environment and over precisely the same timescale (detailed in Section 2 and *Box 1*). In the example of *Figure 1*, collective fitness is computed by considering a full collective generation, while particle fitness is computed within collective development. This renders the comparison spurious, as illustrated in *Figure 2*. Computing the fitnesses at the same timescale and, thus, in the same reference environment would involve simultaneously tallying the increase of individual cells and number of collectives in a time frame spanning several collective cycles.

The export-of-fitness account has been refined by *Shelton and Michod, 2020* in light of the counterfactual approach to the evolution of multicellularity they developed earlier (*Shelton and Michod, 2014*). Following this account, decoupling and/or transfer occur not simply from the particle-level to

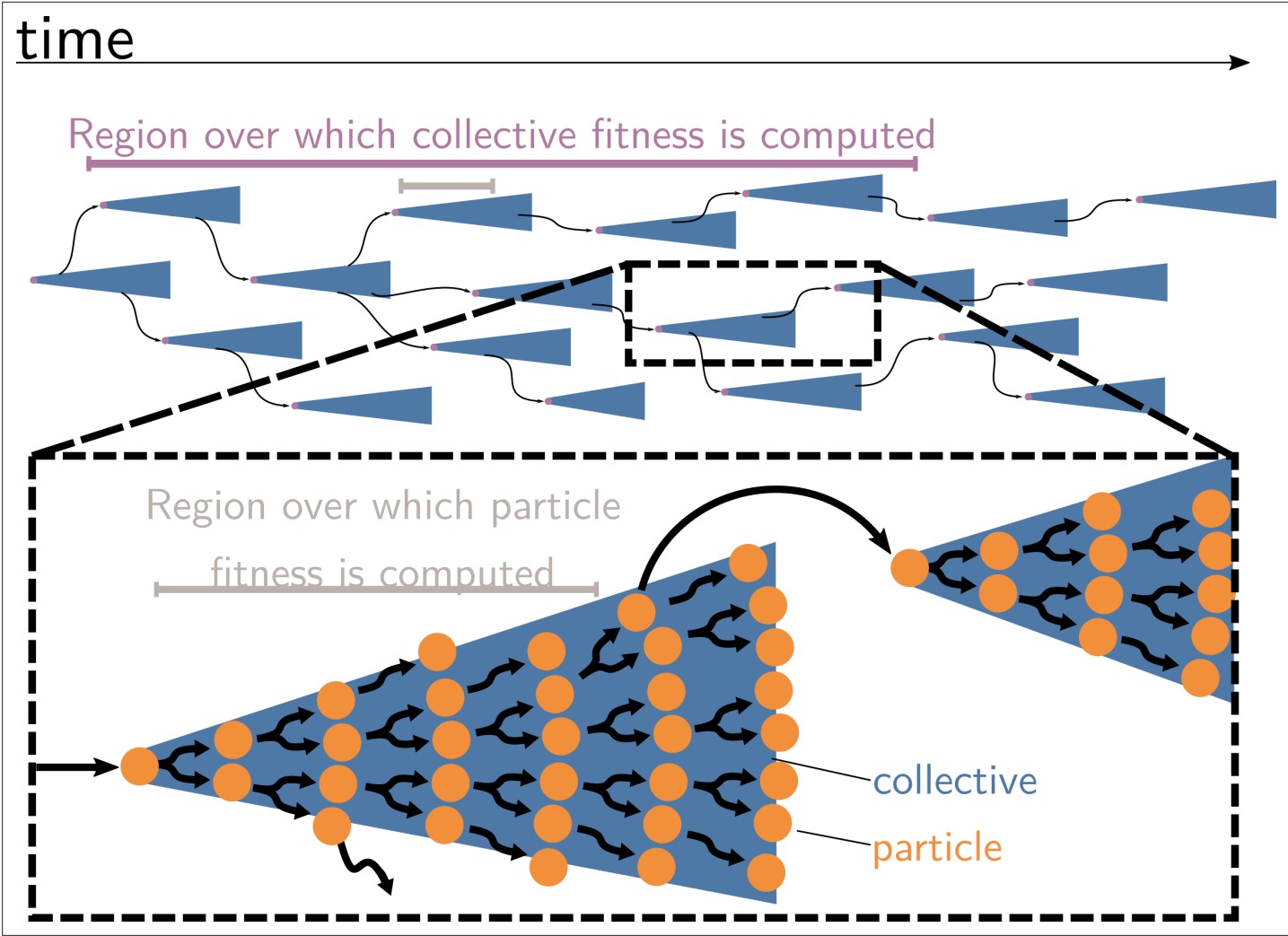

**Figure 2.** Collective and particle fitnesses are not measured over the same environment. During an evolutionary transition in individuality, there are two levels of organisation: collectives (blue triangles) are composed of particles (orange disks). Both levels have their own genealogy (black arrows). Collective fitness is computed by considering one or several full collective generations (purple timeline), while particle fitness is computed within each collective development (grey timeline). As a result, they may exhibit opposing dynamics (increasing for collective and decreasing for particle fitness), giving rise to the fitness-decoupling observation.

the collective-level fitness, but between the *counterfactual* particle level (i.e., particles as they would be if they were not part of the collective) and the collective level (*Shelton and Michod, 2020*). This interpretation alleviates some of the ambiguities present in earlier work by explicitly stating a point that was previously open to interpretation. However, we suggest that it is incompatible with the use of the notion of export of fitness. This is so because the vocabulary suggests that transfer actually occurs between actual and counterfactual quantities, the latter of which do not actually exist by definition.

To summarise, there are two ways in which the export of fitness has been used in the literature: first as a transfer between actual quantities, and second as a process concerning counterfactual quantities (note that since *Shelton and Michod, 2020*, the original authors have explicitly favoured the second interpretation). We suggest that neither account is satisfactory—actual export of fitness is impossible because actual cell and collective fitness are equal, and counterfactual export of fitness is inconsistent because there is no material quantity to be transferred. The first point has significant support within the literature (*Shelton and Michod, 2014*; *Bourrat, 2015a*; *Bourrat, 2015b*) and can be proved for a large class of population dynamics models (as we do in Section 2). The second point is more contentious; however, it primarily concerns the limits of the notion of export of fitness rather than the use of the counterfactual method to study ETIs and detect tradeoffs. In light of these two points,

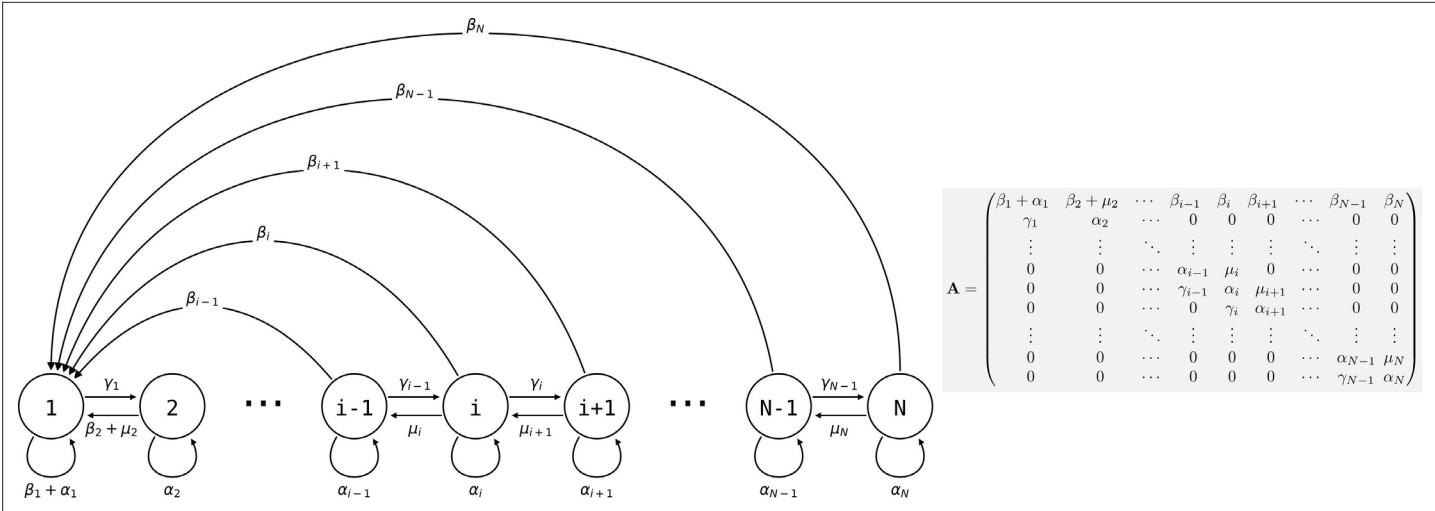

**Figure 3.** Life cycle of collectives as a size–class population projection model. Circles represent a size class of collectives; arrows represent the flow of individuals between size classes. At each time step, collectives of size class $i$ can grow (if $i < N$), shrink, or stay the same size. They also leave propagules of size class 1. See main text for details.

we distinguish between, on the one hand, the notions of 'fitness decoupling', 'fitness transfer', and 'export of fitness', and, on the other, 'fitness-decoupling observations' (see Appendix 1, 'Glossary'). Consequently, we have no objection to the use of fitness-decoupling *observations* in the context of ETIs. To clarify this discussion, Sections 3–5 build upon Section 2 to unfold a general trait-based adaptive scenario of ETIs that is compatible with both experimental fitness-decoupling observations (that stem from a tradeoff) and their deviations.

## 2. Commensurably computed particle and collective fitness are equal

In this section, we present a population projection model (*Leslie, 1945*; *Caswell, 1989*) of an abstract proto-multicellular organism. The population is divided into classes corresponding to different sizes of collectives, as is commonly done in the literature (*Tarnita et al., 2013*; *Pichugin et al., 2017*). We define collective fitness in the context of this model (Section 2.2) and illustrate the different conventions that can be used to define particle fitness (Section 2.3). We provide proof that, under minimal conditions, if particle and collective fitness are computed with respect to the same set of events (i.e., in the same reference environment), they are equal (Section 2.3). Finally, we use this result to clarify conditions under which fitness-decoupling observations can be made (Section 2.4). Significantly, this model assumes that particles and collectives can coexist. In other words, we do not model the potential competition between particles and collectives, as done by *Tarnita et al., 2013*. Also, we do not study the problem posed by the existence of free-riders or cheaters in collectives. However, this could be done by adding a game-theoretic layer to our model.

### 2.1 Modelling a nested demography

Consider a population of genetically homogeneous *particles*, structured into *collectives* following the life cycle illustrated in *Figure 3*. Each collective is characterised by its size—that is, the number of particles it comprises. At each time step, a proportion $\gamma_i$ of collectives of size class $i$ increases to size class $i + 1$, a proportion $\mu_i$ shrinks to size class $i - 1$, a proportion $\delta_i$ dies, and the remaining collectives (i.e., a proportion $\alpha_i = 1 - \delta_i - \mu_i - \gamma_i$) stay the same size. Let $N$ be the maximum size above which collectives cannot grow. Here, the collectives are voluntarily abstract because the statement the model supports is quite general, but one can consider that the particles are cells and growth and shrinkage of the collective is mediated by cell duplication and death. Additionally, collectives reproduce: a collective of size $i$ produces on average $\beta_i$ propagules of size class 1 at each time step. Such a life cycle can be represented as a population projection model, whose dynamics are given by

$$c_i(t+1) = \sum_{j=1}^{N} a_{ij}c_j(t),$$

where $c_i(t)$ is the density of collectives of size $i$ at time $t$, and $a_{ij}$ is the weight of the $i \rightarrow j$ edge of the life cycle graph (**Figure 3**; the $a_{ij}$ are the elements of the population projection matrix $\mathbf{A}$) for all values of $i$ and $j$ between $1$ and $N$. It follows that the population density of particles is given by

$$n(t) = \sum_{j=1}^{N} k_j c_j(t),$$

where $n(t)$ is the number of particles within the population at time $t$, and $k_j$ the number of particles corresponding to size class $j$. For instance, if we consider that all particles reproduce once for the collective to grow one size class, then $k_j = 2^{j-1}$.

## 2.2 Computing collective fitness

We define the fitness of collectives as the Malthusian parameter or asymptotic exponential growth rate of a population of collectives sharing the same trait value (**Metz et al., 1992**). If all transitions represented in the life cycle graph are possible—that is, $0 < \alpha_i < 1 \ \forall i = 1, 2, ..., N-1$, as well as $\beta_i > 0 \ \forall i = 1, 2, ..., N-1$—then the matrix $\mathbf{A} = \left(a_{ij}\right)_{i,j \in [1,2...N]}$ is non-negative and primitive. Following the Perron-Frobenius theorem (see **Caswell, 1989**, p. 57), there exists a positive eigenvalue of $\mathbf{A}$, noted $\lambda$, called the dominant eigenvalue of $\mathbf{A}$, with an associated non-negative eigenvector $\mathbf{w}$. Moreover, the strong ergodic theorem (see **Caswell, 1989**, p.57) shows that the long-term dynamics of the population are described by a growth rate and a stable population structure $\mathbf{w}$. When these assumptions are met, the fitness of collectives is given by:

$$F = \lim_{t \to \infty} \left( \frac{ln\left(\sum_{j=1}^{N} c_j(t)\right) - ln\left(\sum_{j=1}^{N} c_j(0)\right)}{t} \right) = ln(\lambda) \tag{1}$$

where the first equality is given by the definition of fitness, the second by the Perron–Frobenius and strong ergodic theorems, and $c_j(t)$ is the density of collectives of size class $j$ at time $t$.

## 2.3 Computing particle fitness

In contrast to collective fitness, computing particle fitness is more challenging. There are, prima facie, at least three ways to compute the fitness of a particle. Each gives rise to a different measure that we will call $f_1$, $f_2$, and $f_3$ (illustrated in **Figure 4**; their expression for the current model will be given later). To compute:

- $f_1$, look within each collective and consider the dynamics of the particles. This is equivalent to ignoring all the 'between collective' level events (collective births and deaths). This is what is done experimentally when measuring cell density within isolated collectives (**Figure 1**, **Hammerschmidt et al., 2014**).
- $f_2$, look at a theoretical monoparticle collective. This is equivalent to ignoring all the 'within collective' events (collective growth and shrinking). This conception of fitness corresponds to the notion of 'counterfactual fitness' proposed by **Shelton and Michod, 2014**; **Shelton and Michod, 2020** because it is equivalent to the fitness that particles *would* have if they were genetically equivalent (same trait values), but without the ability to produce multiparticle collectives. This is also close to the experimental measurements of cell density performed experimentally (**Hammerschmidt et al., 2014**; 'growth rate' in ED **Figure 4**) under conditions preventing the formation of collectives (shaking).
- $f_3$, take into account all events (i.e., within and between collective), with counts of the number of particles through time.

Each of these ways to compute fitness uses a different reference environment (including the timescale). Each can be adequate in different contexts. It is appropriate to measure particle fitness

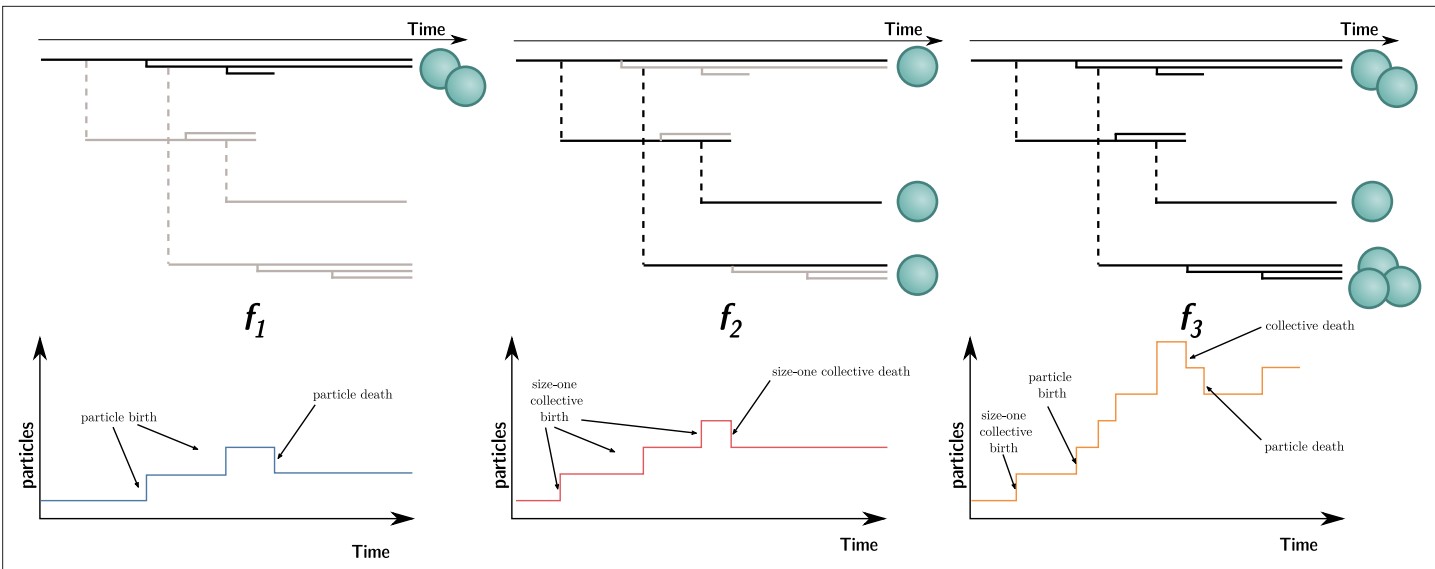

**Figure 4.** Three ways to compute particle fitness ($f_1$, $f_2$, and $f_3$) in a lineage starting from a single particle. Top: each solid horizontal line represents the life span of a particle. The vertical axis has no unit and only represents population structure. Particles within the same collectives are represented as a series of horizontal lines close to one another (isolated particles count as monoparticle collectives). Each dashed vertical line connects a parental collective to its offspring (which starts as a single cell). Greyed-out elements for $f_1$ and $f_2$ indicate the processes that are omitted during the fitness calculation. Bottom: the time series below each example shows how particles are tallied. In the case of $f_1$, between-collective events are ignored. Thus, the two particles are the only ones relevant for its computation, whereas, in the case of $f_2$, within-collective birth–deaths and their effects are ignored. Thus, the three particles (i.e., monoparticle collectives) are relevant for its computation. Finally, in the case of $f_3$, the full lineage is used. Thus, all six particles must be taken into account for its computation.

considering within-collective events ($f_1$) only when the evolutionary process studied occurs in the short term. For instance, computing $f_1$ can tell us which mutant cell lineage can take over within an organism (e.g., cancerous lineages). The counterfactual method $f_2$ gives information about a 'what if' world where particles cannot be organised in collectives (i.e., in the model, collectives cannot grow: $\gamma_i = 0$). It might give information on the unicellular ancestor of a collective. Indeed, a reasonable hypothesis is that the ancestral trait values are those that maximise counterfactual fitness (**Shelton and Michod, 2020**, p. 8).

However, there is no a priori reason for the values of $f_1$, $f_2$, and $f_3$ to be equal to each other, or even to change in the same direction when the traits of the organism change. This is clear in view of their expression in the model. Considering within-collective dynamics (ignoring the effect of density dependence, as is usually done in experiments) gives $f_1 = \ln(1 - \mu_1 + (k_2 - 1)\gamma_1)$. (That is, the expected exponential growth rate of a discrete-time branching process (or Bienaymé–Galton–Watson process) where each individual leaves either 0, 1, or $k_2$ offspring with probabilities $\mu_1$, $(1\mu_1)(1g_1)$, and $(1\mu_1)g_1$, respectively.) In contrast, considering the counterfactual collective dynamics gives $f_2 = \ln(1 - \delta_1 + \beta_1)$. Comparing the two equations, we can see that $f_2$ does not depend on $\gamma$. It follows that an increase in $\gamma$ would result in an increase in $f_1$ with no consequence for $f_2$. *Mutatis mutandis*, the same applies for $\mu$, $\delta$, and $\beta$.

The case of $f_3$ is different. The expression of $f_3$ is given by the long-term exponential growth rate of $n$, which is a linear combination of exponentially growing terms $k_i c_i(t)$ and, thus, asymptotically grows at the rate of its largest coefficient. By definition, this coefficient is $\lambda$, the dominant eigenvalue of $\mathbf{A}$. Intuitively, this means that once the population has reached a stable size distribution, the relative proportion of different collective sizes does not matter for computing the overall growth rate of particles—they grow at the same rate as the collective population. Thus, $f_3 = \ln(\lambda) = F$. It follows that a change in $\gamma$ (or $\mu$, $\delta$, and $\beta$) would affect $f_3$ and $F$ in the same way. This shows that when the particle fitness is computed with the same frame of reference (i.e., the same environment) as collective fitness (using the same timescale and the same set of events), the values are (mathematically) equal.

## 2.4 Counterfactual fitness-decoupling observations can be useful to detect tradeoffs

With these distinctions in place and the constraint that, to be compared, fitnesses must be measured over the same set of events (i.e., same environment over the same timescale; see *Box 1* and Section 1), the apparent contradiction of a simultaneous increase in collective fitness and decrease in particle fitness of a single biological entity is dissolved. More precisely, either fitness at the particle level and the collective level are commensurably computed—that is, with respect to the same biological object and in the same reference environment—in which case they are equal ($F$ and $f_3$), or they refer to different biological settings ($f_1$ or $f_2$ and $F$) and, thus, the biological significance of their differential dynamics is not immediately clear.

While these points have been made in the literature (see Section 1), our mathematical proof applies to a larger class of population dynamics models—it is not tied to an MLS1/MLS2 framework and does not require any assumptions regarding the relationship between particle and collective life history traits, such as the usual assumption that collective viability is a linear function of individual viability (since the $\gamma, \mu, \delta, \beta$ are free). It only requires that the population reaches a stable size distribution; only if collectives are able to grow (or shrink) indefinitely, which is not a realistic assumption for ETIs, could genuine fitness decoupling be observed (see *Bourrat, 2021a*, Chapter 5 for details).

Although observations that $f_1$ or $f_2$ decrease over time do little to clarify the process of an ETI—a point to which we return in Section 3.3—such observations can be understood as a consequence of an underlying tradeoff and could, thus, prove useful in their experimental assessment. In the next section, we introduce a model of such a tradeoff that provides a plausible biological mechanism for the emergence of multicellular collectives under given conditions, while displaying a simultaneous increase in $F$ and a decrease in $f_2$. However, tradeoffs leading to this observation might not exist or be detected in all ETIs. We present such a biological scenario in Section 5.

## 3. Tradeoffs between particle traits drive ETIs

The previous section showed that if collective and particle fitness are computed in the same reference environment, they are necessarily equal. In addition, it showed that a fitness-decoupling observation (e.g., $f_2$ and $F$ going in different directions) offers little information about the mechanism of ETIs on its own since there is no reason to relate the dynamics of $f_1$ or $f_2$ to that of collective fitness $F$ without additional assumptions about the system (i.e., that ancestral particles and particles within the collective are restricted to the same subset of the trait space, e.g., by a tradeoff). In this section, we present an evolutionary model of ETIs that includes these additional assumptions (Section 3.1), describe under which conditions fitness-decoupling observations can occur (Section 3.2), and provide an example involving a tradeoff between two traits: the survival of existing collectives and the production of new cells (Section 3.3). This example features an increase of $F$ and a decrease of $f_2$ along the evolutionary trajectory. We focus on $f_2$ because it is more relevant to ETIs; while $f_1$ has been discussed more strictly in the levels of selection literature, $f_2$ can be interpreted as the hypothetical fitness an ancestor cell would have (*Shelton and Michod, 2020*). Nonetheless, the same argument could be made for $f_1$.

### 3.1 Modelling evolution

The demographic model of Section 2 is completed with a model of evolution in two steps. First, consider that the life cycle of collectives (summarised by the population projection matrix **A**) depends upon a trait $\theta \in [0, 1]$ whose value can change by mutation. We consider here mutations to be abstract phenotypic changes—the mechanism by which they are produced could be either phenotypic plasticity, environmental change, or actual genetic mutations. For each trait value $\theta$, a corresponding fitness value $F(\theta)$ exists. Second, we use the simplifying assumptions of Lande's Equation (*Caswell, 1989*, p. 164)—namely, the separation of demographic and evolutionary timescales—and the absence of density-dependent effects, and perform a successive-invasion analysis. These assumptions lead to the following equation for the evolution of the average trait value $\bar{p}$ in the population:

$$\frac{d\bar{\theta}}{dt} = \sigma_\theta F(\bar{\theta})^{-1} \left. \frac{dF}{d\theta} \right|_{(\theta = \bar{\theta})}, \tag{2}$$

where $\sigma_\theta > 0$ is the variance of mutation effects on the trait $\theta$. As a consequence, the model predicts that the average value of the trait $\bar\theta$ will 'climb up' the fitness gradient $\frac{dF}{d\theta}$ as fitter mutants emerge and invade the population.

The ancestral phenotype (i.e., the initial condition $\bar\theta(t = 0)$ of the evolutionary trajectory) will be taken to be $\theta_0$, the optimal trait value for the ancestral unicellular organism. Thus, $\theta_0 = \arg\max_\theta f_2(\theta)$, where $f_2(\theta)$ is the value of $f_2$ for an organism with trait $\theta$. Similarly, let $\theta^*$ be the optimal trait value for the collective, where $\theta^* = \arg\max_\theta F(\theta)$. The code implementing this model and drawing the figures presented here is available as supplementary material (doi: 10.5281/zenodo.5352208). It uses the Matpopmod library (*Bienvenu and Doulcier, 2021*).

## 3.2 Conditions for fitness-decoupling observations between $f_1$ or $f_2$ and $F$

Once a mutation (or change in the environment) makes the first multiparticle collective possible, the process that drives population dynamics is one associated with $F$. A sufficient condition for fitness-decoupling observations to be possible would be that the trait values that are optimal for monoparticle collectives ($\theta_0$) are not optimal for multiparticle collectives ($\theta^* \neq \theta_0$), provided the fitness landscape is sufficiently smooth (and both $F$ and $f_2$ are continuous functions of $\theta$ with a single maximum value). Thus, as selection acts during the ETI and drives the trait values towards fitter collectives (towards $\theta^*$, the optimal value of $\theta$ for $F$), this would necessarily lead to less fit 'counterfactual' monoparticle collectives (away from $\theta_0$, the optimal value for $f_2$). Conversely, a sufficient condition to observe 'coupling' between $f_2$ and $f_3$ (or $F$) would be that the optimal trait values for these two measures coincide ($\theta^* = \theta_0$) and that the ancestral trait was not optimal. In such a condition, the evolutionary trajectory of the population would tend towards traits with a higher value of $F$ and, coincidently, towards higher values of $f_2$ (hence the existence of a 'coupling').

## 3.3 An example: The tradeoff between collective growth and persistence

Let $\theta \in [0, 1]$ be a trait that controls the relative investment of the particles towards collective survival and collective growth, modelled by parameters $s$ and $b$, such that $s = \theta$ and $b = 1 - \theta$. More precisely, let the probability for a collective of size class $i$ to survive a single time step be $p_i$, with $p_1 \in [0, 1]$ and for $i = 2, \ldots, N$, $p_i = 1 - e^{-\eta s}$ (where $\eta$ is a scaling factor). Let the probability for a collective of size class $i$ to grow to the next class size during a time step be $g_i$, with $g_i = 1 - e^{-\eta b}$ for $i = 1, \ldots, N-1$ and $g_N = 0$. Additionally, let the expected number of propagules shed by a collective of class size $i$ be $m_i = \eta b k_i$. Thus, following the Birth-Flow Class structured model (*Caswell, 1989*, pp. 83–93), the matrix projection model from *Figure 3* is parameterised as such: $\alpha_i = p_i(1 - g_i)$, $\gamma_i = p_i g_i$, and $\beta_i = p_1^{0.5} \frac{1}{2}((1 + \alpha_i)m_i + \gamma_i m_{i+1})$. For this example, consider that collectives cannot shrink by setting $\mu_i = 0$.

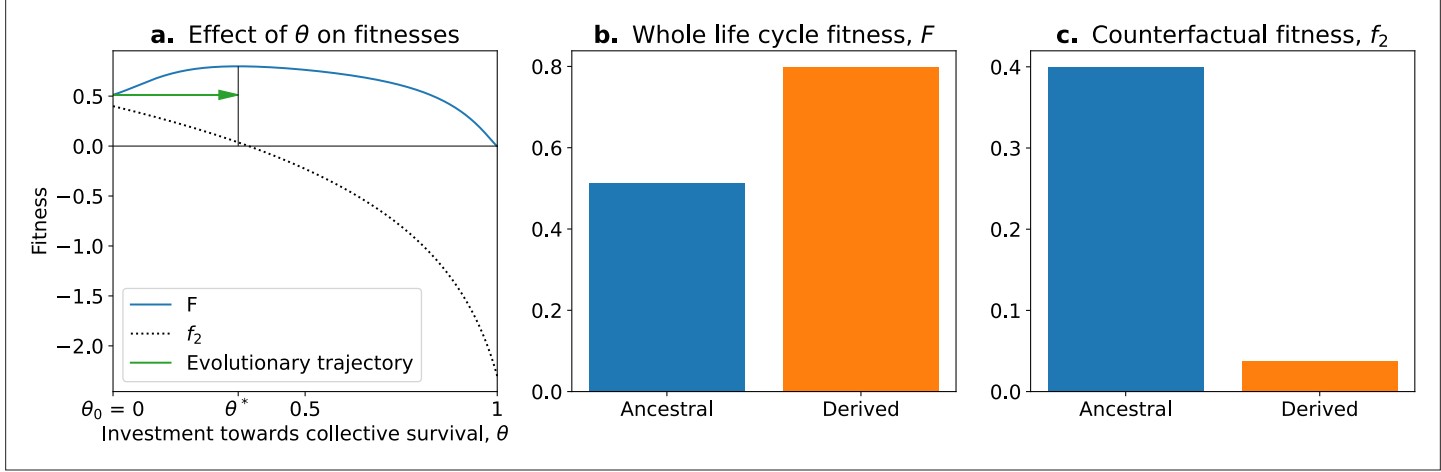

**Figure 5.** The tradeoff model can reproduce the fitness-decoupling observation. (**a**) Values of $F$ and as a function of the trait $\theta$. (**b**) Ancestral and derived values of whole life cycle fitness ($F$). (**c**) Ancestral and derived values of counterfactual fitness ($f_2$). The expected evolutionary trajectory from ancestral ($\theta_0$) to derived ($\theta^*$) trait value (green arrow in (**a**)) results in an increase of $F$ and a decrease of $f_2$, reproducing the fitness-decoupling observation of *Figure 1*. Parameters: $N = 30$, $\eta = 8$, and $p_1 = 0.1$.

In this model, the optimal trait value for counterfactual ($f_2$) fitness is always a null investment in collective survival $\theta = 0$. However, the optimal trait value for whole life cycle fitness ($F$) is $\theta > 0$. Thus, if a population starts with the optimal trait value for the counterfactual fitness $\theta_0 = 0$, it will evolve towards the optimal value $\theta^* > 0$ (*Figure 5*; green arrow). Over time, collective-level fitness $F$ increases while counterfactual fitness $f_2$ decreases.

This evolution towards higher values of $\theta$ (because it increases $F$) coinciding with a decrease in $f_2$ leads to a fitness-decoupling observation. Note, importantly, that following the assumptions of the model, the opposite directions of the dynamics of $F$ and $f_2$ are a consequence of optimal trait values constrained by a tradeoff being different in different environments, rather than an inherent relationship between them. To see this, consider the example of selection for fast settlement in *Saccharomyces cerevisiae* (*Ratcliff et al., 2012*). The optimal trait value for an ancestral particle (i.e., a free living cell) is having round cells. However, in a snowflake aggregate, the optimal shape is to be elongated because this shape permits packing of more cells. In this example, $f_2$ decreases—to be elongated in the ancestral environment would be disadvantageous—when whole life cycle fitness $F$ increases: to be elongated is advantageous in the context of the snowflake. Suppose for an instant that elongated cells were also favoured in the unicellular context, but the initial conditions of the dynamics were still round cells: the evolutionary trajectory in a collective context would not be modified. The value of the counterfactual fitness, in and of itself, has no consequence for the outcome of selection in the collective context: it is not because $f_2$ decreases (and is transferred to $F$, or is decoupled from $F$) that the transition occurs. The trait values evolve according to $F$. There is no direct causal relationship between $f_2$ and the observed changes in traits. Relative changes in $F$ and $f_2$ do not cause the dynamics observed but rather follow from changes in environmental conditions and genetic and phenotypic constraints on cells. Nevertheless, we consider the counterfactual fitness approach (*Shelton and Michod, 2014*) a valuable tool to detect tradeoffs between traits in the context of ETIs. This approach can also be used to generate hypotheses about the traits of the unicellular ancestor by considering that ancestral traits had an optimal value for counterfactual fitness.

In the next section, we propose to go one step further and evaluate how the capacity for a lineage to break away from such a tradeoff could be used to detect an ETI. Patterns of tradeoff breaking correspond with the emergence of novel collective-level traits—that is, traits that can only be exhibited in a collective context—and, as such, provide an evolutionary cause of ETIs.

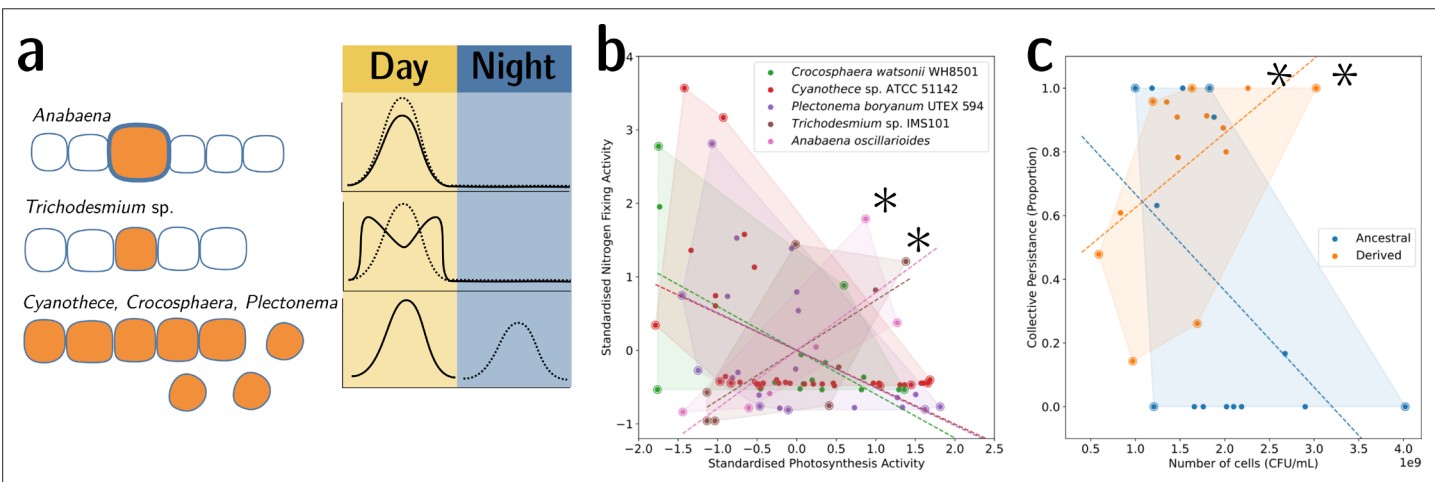

**Figure 6.** Tradeoff-breaking lineages can be inferred experimentally. (**a**) Morphological and physiological N₂-fixing adaptations for different cyanobacteria. Orange shaded areas indicate nitrogenase localisation. Daily rhythm of photosynthesis (solid line) and N₂-fixation (dashed line) (modified from Figure 2 of *Berman-Frank et al., 2003*). (**b**) Tradeoff between photosynthesis activity and nitrogenase activity in cyanobacteria (data taken from *Colón-López et al., 1997*; *Mohr et al., 2013*; *Misra and Tuli, 2000*; *Berman-Frank et al., 2001*; *Popa et al., 2007* and standardised). The shaded area for a given species corresponds to the convex hull of observations. Assuming a representative sampling, it stands for the expected range of traits accessible for this species. Dashed lines are least-square linear regressions of the observations of each species; asterisks indicate potential tradeoff-breaking observations because they depart notably from the tradeoff pattern displayed by most species. (**c**) Tradeoff between collective persistence and cell number in *Pseudomonas fluorescens* for ancestral and derived lineages (*Hammerschmidt et al., 2014*; asterisks indicate tradeoff-breaking observations in two lineages). Dataset published as *Rose et al., 2018*.

## 4. Tradeoff breaking as a marker of an ETI

In the previous section, we saw that a tradeoff between traits can result, under specific conditions, in an empirical fitness-decoupling observation. Evolutionary tradeoffs between traits are a consequence of the genetic background of organisms and their environment. Thus, they are not immutable and can evolve if some changes in the genetic background or the environment occur. In this section, we propose that a marker of an ETI is 'breaking' from the initial tradeoff (hereafter called 'tradeoff-breaking' observations). To explain this phenomenon, we present a modification to the model described in Section 3, which we call the 'tradeoff-breaking model'.

A tradeoff is essentially a constraint on the combined values of a set of traits; it prevents a given organism from simultaneously performing well in two or more functions—for instance, growth and survival or photosynthesis and nitrogenase activity in cyanobacteria, as discussed below. Thus, if a mutant lineage is able to bypass the tradeoff and perform well in different functions, it is expected to be fitter than its ancestor and increase in proportion in the population. In some cases, mutations that lead to a multicellular morphology might be a necessary step towards bypassing or breaking the tradeoff. This provides an 'adaptive' explanation for the emergence of multicellular organisms. Collectives (i.e., multicells) emerge by mutations constrained by the tradeoff, but their long-term persistence is rendered more probable by tradeoff-breaking mutations that would not have been possible, or would not have broken the tradeoff, had they occurred in their unicellular ancestors.

To illustrate this point, we provide two examples from biology. The first is the well-understood tradeoff between photosynthesis and nitrogen fixation—from dinitrogen gas ($N_2$)—in cyanobacteria. The tradeoff is caused by the oxygen sensitivity of the enzyme nitrogenase, which catalyses the process of reducing $N_2$ to ammonia ($NH_3$). This prevents cells from performing both functions simultaneously and has resulted in several morphological and physiological adaptations (*Figure 6a*). In the unicellular species, *Cyanothece* sp. ATCC 51142 and *Crocosphaera watsonii* WH8501, and the undifferentiated filamentous *Plectonema boryanum* UTEX 594, the two functions are separated temporally by a circadian rhythm: the oxygen-sensitive $N_2$ fixation is performed during the night, unhindered by the oxygen-producing photosynthesis during the day. When plotting the activity values for photosynthesis and $N_2$ fixation for populations of these species, they fall on both sides of the tradeoff—depending on the time of the day (*Figure 6b*). In the morphologically undifferentiated filamentous multicellular *Trichodesmium* sp. IMS101, the two functions are performed simultaneously but in different, morphologically identical cells of the filament. Thus, the values for these populations are located in the middle of the tradeoff—they perform averagely in both functions. This pattern can also be seen for populations of the highly differentiated filamentous *Anabaena oscillarioides*, which perform even better than the undifferentiated *Trichodesmium* sp. This can be explained by the presence of differentiated cells (heterocysts) that only fix $N_2$ and exchange the fixed nitrogen compounds against carbon products with the photosynthesising cells of the filament. This example seems to be compatible with the tradeoff-breaking framework—both multicellular and (physiologically) differentiated species, (a) *Anabaena oscillarioides* and *Trichodesmium* sp., seem to have broken away from the tradeoff, which leads to tradeoff-breaking observations in *Figure 6b* (indicated by asterisks). Moreover, photosynthesis and nitrogen fixation are positively associated, as indicated by the regression line. Overall, the example of cyanobacteria illustrates how tradeoff breaking can occur—multicellular differentiated morphology and designated $N_2$-fixing cells allow the organisms to break away from the tradeoff that is present in the unicellular and physiologically undifferentiated phyla (visible in *Figure 6b* as the three negative regression lines).

The second example is the derived *P. fluorescens* populations, where tradeoff breaking seems to have occurred in some of the evolved lineages. In this case, the tradeoff implies that collective persistence cannot increase concomitantly with cell number. However, in the experiment of *Hammerschmidt et al., 2014*, two lineages have succeeded in doing this, leading to two tradeoff-breaking observations visible in *Figure 6c* (indicated by asterisks). Here, collective persistence increased due to the evolution of a *mutS*-dependent genetic switch that enabled rapid and predictable transitioning between two stages of a life cycle. This increase in collective persistence is not accompanied by a decrease in cell density, as is the case in other lineages, indicative of these two lineages having broken the tradeoff.

The model presented in Section 3 cannot account for such changes affecting the traits, for example, through mutations. To do so, it must be modified into what we refer to as the tradeoff-breaking

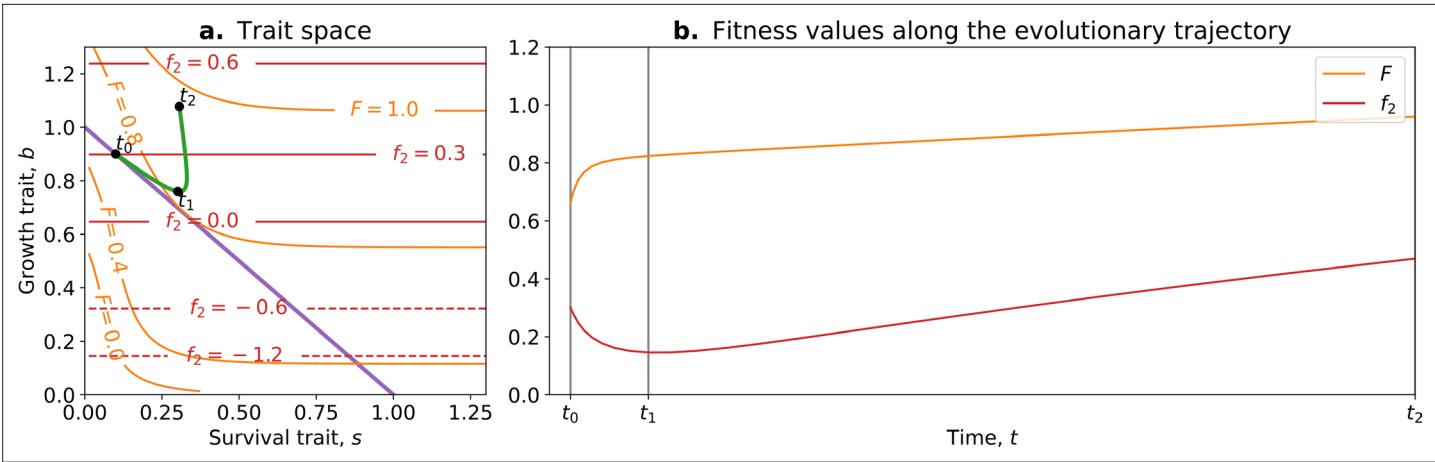

**Figure 7.** Tradeoff-breaking mutations do not fit the fitness-decoupling observation. (**a**) Trait space with isolines of fitness. An example of a possible evolutionary trajectory is shown in green. (**b**) Particle (counterfactual; $f_2$ in red) and collective fitness ($F$ in orange) values along the example evolutionary trajectory (in green). The strict tradeoff from Section 3 and **Figure 5** is shown in purple. The times marked by vertical lines in (**b**) correspond to the dots in (**a**). The evolutionary trajectory can be separated into two phases once collectives have been formed: a fast-paced phase (before $t_2$) that closely follows the purple tradeoff in (**a**) and a slower phase (after $t_1$) that breaks away from it and leads to the tradeoff-breaking observation. Note that a fitness-decoupling observation can only be made in the fast-paced phase of the trajectory (before $t_2$), as represented in (**b**). Parameters: $N = 15$, $p_1 = 0.1$, $\eta = 8$, $\rho = -0.9$.

model. In the model presented in Section 3, the traits survival $s$ and growth $b$ were linked by a deterministic relation through the investment trait $p$: $s = p$ and $b = 1 - p$. Thus, any mutation affecting one trait necessarily leads to opposite effects on the other trait. This assumption can be relaxed, allowing the two traits to take any pair of values. To keep modelling the tradeoff, we suppose that the distribution of mutational effects (the mutation kernel) is a two-dimensional Gaussian distribution with a high (negative) correlation value ($\rho = -0.9$). This means that most mutations that increase one trait value reduce the other. However, and this is crucial, some rare mutations have the effect of increasing or decreasing both values, something that was impossible in the model described in Section 3. Thus, Eq. 2 becomes

$$\frac{d}{dt} \begin{pmatrix} s \\ b \end{pmatrix} = F(s,b)^{-1} \begin{pmatrix} \sigma_s^2 & \sigma_s\sigma_b\rho \\ \sigma_s\sigma_b\rho & \sigma_b^2 \end{pmatrix} \nabla F(s,b), \tag{3}$$

where $F(s,b)$ is the whole life cycle fitness as a function of the traits, $\nabla F(s,b)$ is the fitness gradient in the two-dimensional trait space, and $\sigma_s$ and $\sigma_b$ are the variance of mutational effects on $s$ and $b$, respectively. **Figure 7** shows the trajectory resulting from this model. Initially, the population moves along the tradeoff in the trait space: reducing the value of $b$ and increasing the value of $s$. This reflects how 'low-hanging fruit' mutations—mutations that are more frequent due to the skew in the mutational effect distribution—drive the initial dynamics. This fast-paced phase of the dynamics (which can be absent in some cases; see next section), following a phase during which collectives are formed, ends when the population reaches the neighbourhood of the optimal organism that lies on the tradeoff line $s = (1 - b)$ (around $t_1$). Then, a slower phase of the dynamics starts and leads to a tradeoff-breaking observation (the population 'breaks away' from the tradeoff line; **Figure 7a** after $t_1$). This third phase is slower because mutations that move collectives in this direction are statistically less likely to occur. **Figure 7b** shows that observation of fitness decoupling between $f_2$ and $F$ could only be made in the first part of the trajectory. However, the whole trajectory is characterised by an increase in fitness $F$, as discussed in Section 3.

Our model gives a simple mechanism that can reproduce the experimental fitness-decoupling observation, in addition to the tradeoff-breaking observations due to rare mutants. Specifically, we observe that the trajectory resulting from the model is mirrored by the trajectory of lineages in the evolved *P. fluorescens* populations (**Hammerschmidt et al., 2014**). When compared to the ancestral

lineages, most of the evolved lineages appear to have been constrained by the tradeoff and increased in collective persistence at the expense of cell density (through low-hanging fruit mutations, as is the case between $t_0$ and $t_1$ in *Figure 7*). Notably, only the two outlier lineages (marked by asterisks in *Figure 6c*) seem to have started to break away from the tradeoff—they seem to have reached the slower phase of the evolutionary dynamic (where less likely mutations are explored, as is the case between $t_1$ and $t_2$ in *Figure 7*). Tentatively, the fact that only the two outlier lineages broke away from the tradeoff might be due to a higher mutation rate increasing the relative speed of evolution compared to the other lines. According to our tradeoff-breaking framework, the fitness-decoupling observation should be reinterpreted as lineages being in the fast-paced phase of the trajectory. Moreover, the tradeoff-breaking lineages should be explored (as done to an extent in *Hammerschmidt et al., 2014*) as we advocate that such tradeoff-breaking observations in this context are the mark of significant innovation and, thus, can be used to detect ETIs.

The tradeoff-breaking model presented here is compatible with a number of different models of ETIs recently proposed in the literature. Among them are two models we think could benefit from being seen through the framework we have developed here because they illustrate a diversity of mechanisms that yield initial tradeoffs and tradeoff-breaking mutations: the ratcheting model proposed by Libby and collaborators (*Libby et al., 2016*; *Libby and Ratcliff, 2014*) and the ecological scaffolding model proposed by *Black et al., 2020*. Further, recasting these two models in terms of tradeoff breaking yields new insights, which are detailed in *Box 2* and *Box 3*, respectively.

In the ratcheting model, proto-multicellular organisms are in an environment alternating between multicell-favouring and unicell-favouring. This yields a tradeoff between the two states and the selection for a high probability of multicells to revert to unicells. Some mutations ('ratcheting (type 1) mutations') are assumed to be beneficial in a collective (multicellular) environment while deleterious in a unicellular context; thus, they play the tradeoff-breaking role. *Libby et al., 2016* showed, through simulations, that the accumulation of ratcheting mutations makes it harder for a multicellular organism to revert to a unicellular state even when the environment becomes favourable for unicellularity.

In the ecological scaffolding model, the environment, structured both spatially and temporally, allows for selection of collective-level properties without the need to assume anything about the particles other than that they reproduce at different rates. *Black et al., 2020* showed that the tradeoff stems from population structure: reductions in cell growth rate are favoured due to benefits to collectives that are realised via improvements in dispersal. The emergence of specialised soma cells is an example of tradeoff breaking: it allows an increase in collective dispersal without requiring as much cell growth reduction.

Both models illustrate the flexibility of our tradeoff and tradeoff-breaking approach. First, it allows multiple mechanisms of evolutionary transitions to be formalised in a unified way. Second, tradeoff-breaking observations can be used as a general marker across various mechanisms of evolutionary transitions.

## 5. The trait-based approach in the context of ETIs

The trait-based approach we have described so far delineates an adaptive scenario for an ETI. This scenario is divided into several phases. These are, first, the formation of collectives, in which individuation mechanisms that might be adaptive are at play (*Clarke, 2013*; *Clarke, 2014*); second, an optimisation within the constraints of the ancestral tradeoff (note that this phase is optional and might be shortened or bypassed depending on the biological system); and third, a tradeoff-breaking phase. In this section, we recast this scenario in the broader context of ETIs and discuss its relevance as a marker of individuality.

Before an ETI starts, the unicellular ancestors have been selected for traits that optimise growth rate in a unicellular environment under the constraints of their genetic constitution ('Tradeoff-optimal particle phenotype'; blue disk in *Figure 8*). Then, a mutation (e.g., loss of the transcription factor ACE2, resulting in snowflake-shaped yeast clusters; see *Ratcliff et al., 2015*), a plastic change in phenotype (e.g., filament formation under low population densities in cyanobacteria; see *Tang et al., 2022*), a change in the structure of the environment (as in the ecological scaffolding model for the origin of multicellularity; see *Black et al., 2020*), or even a combination of several factors ('wrinkly spreader' mats arising by mutation and ecological scaffolding; see *Hammerschmidt et al., 2014*) promote the formation of collectives (*Figure 8*). Multicellular collectives define a new environment

## Box 2. Ecological scaffolding and tradeoff breaking.

Population structure can lead to the kind of tradeoff presented in Section 3, as seen in the ecological scaffolding scenario for the origin of multicellularity (**Black et al., 2020**). In this scenario, the population of particles is structured in patches of finite resources with dispersal between patches. The tradeoff in this model is between trait values that enhance particle performance within patches and trait values that favour dispersal to new patches. The evolutionary dynamics of two particular traits are studied: particle growth rate and production of soma-like particles that do not disperse themselves but favour the dispersal of the other particles in the patch. In this box, we show how the model of **Black et al., 2020** captures the concepts of tradeoff and tradeoff breaking presented in the main text.

### Ecology

We model the dynamics of germ cells ($g$), soma cells ($s$), and resources ($r$) within collectives. Two traits can mutate: the growth rate of germ cells $\beta$ and the proportion of soma cells that are produced by germ cells $q$. The ecology within a patch is given by (Equation 3 in **Black et al., 2020**)

$$\begin{cases} \frac{dg}{dt} = & N^{-1}\beta(1-q)r(t)g(t) - g(t) \\ \frac{dr}{dt} = & -N^{-1}\beta r(t)g(t) - dr(t)s(t) \\ \frac{ds}{dt} = & N^{-1}\beta q r(t)g(t) - s(t) \end{cases},$$

where $N$ is the carrying capacity of a patch and $d$ the rate at which soma cells consume resources. Initial conditions at the beginning of each generation are taken to be $g(0) = 1$, $r(0) = N$, and $s(0) = 0$. Thus, at the beginning of a collective generation, there is only a single germ cell in the collective. Note that if $q = 0$, there are never any soma cells in the model (for any point $t$, $s(t) = 0$).

The weight of a patch $w$ in the dispersal phase is given by $w = (1 + \rho s(T))g(T)$, where $T$ is the duration of the growth phase and $\rho$ is the advantage in dispersal conferred by the soma cells. If $\rho = 0$, the soma cells do not affect the dispersal.

### Fitnesses

We now recast this model within our framework. The within-collective fitness of cells $f_1$ (ignoring inter-collective events and density dependence within collectives) is $f_1 = \frac{d(s(t)+g(t))}{dt}\Big|_{s,r,g=(1,N,0)} = \beta$. The counterfactual fitness $f_2$ is computed, assuming that collectives give rise to free living cells at rate $\beta$ (and not allowing the production of soma cells); thus, $f_2 = \beta$.

Computing the whole life cycle fitness $F$ (or $f_3$) is more challenging since there is (some) density dependence between collectives. However, since collective generations are non-overlapping and collectives only reproduce once (at the end of their life), the only number that matters, in the long run, is the weight of a patch (the number of dispersing propagules) at the time of dispersal: $F \approx w$.

**Box 2—figure 1a** shows the set of accessible phenotypes when mutations occur on either $\beta$ or $q$. From this figure, it is possible to predict what will happen in a hypothetical scenario of sequential mutations on $\beta$ then $q$. Starting from point $a$, if only $\beta$ is able to mutate, we can expect the population to move to $b$ (the highest value of $F$ for $q = 0$, while $f_2$ decreases), optimising on the tradeoff. Note that $\beta$ plays the same role as $\theta$ in Section 3, encoding the linear abscissa of the cells on the tradeoff line. If mutations affecting $q$ become possible once

the population reaches $b$, the population is expected to evolve towards $c$, without change in $f_2$, breaking the previously defined tradeoff. This requires that mutations happen on one trait and then the other. This assumption will be relaxed in the next paragraph by using the same method as in Section 4.

### Evolution

Let $(q, \beta)$ be the vector of traits characterising a phenotype. We compute the evolutionary trajectory using the same Lande equation model as in Section 4: namely, Eq 3. $G$ is the variance–covariance matrix of mutational effects. In this example, consider that mutational effects on both traits are not correlated ($\rho = 0$, using the notation of the main text), and that the mutational effect variance for $\beta$ is much higher than for $q$: $\sigma_\beta = 0.1$ and $\sigma_q = 0.001$. This assumes that most mutations have a higher effect on $\beta$ than $q$.

**Box 2—figures 1b and c** show a trajectory simulated this way, with initial conditions $(q_0, \beta_0) = (0, 1.8)$, which displays a dynamic akin to the one in Section 4: a fast-paced phase where frequent low-hanging fruit mutations (mainly affecting $\beta$) are reached by the population, increasing $F$ (between $t_0$ and $t_1$, note that $f_2$ simultaneously decreases). This fast-paced phase is followed by a slow-paced phase where an increase in $F$ is only possible through rarer mutations (mainly affecting $q$) (after $t_1$; note that $f_2$ simultaneously increases) and leads to a tradeoff-breaking observation similar to the one described in Section 4.

To summarise, this simple set of hypotheses (initial conditions, rarer mutations on $q$ than on $\beta$) leads to a transient fitness-decoupling observation. Notably, this observation does not stem from mutational effects (like in the main text), but from the ecological constraints on $\beta$ that do not allow $F$ and $f_2$ to be maximised for the same conditions (purple line in **Box 2—figure 1a**). Tradeoff breaking is due to rarer mutations on $q$ (as in the main text where rare mutations increase both survival and growth rate).

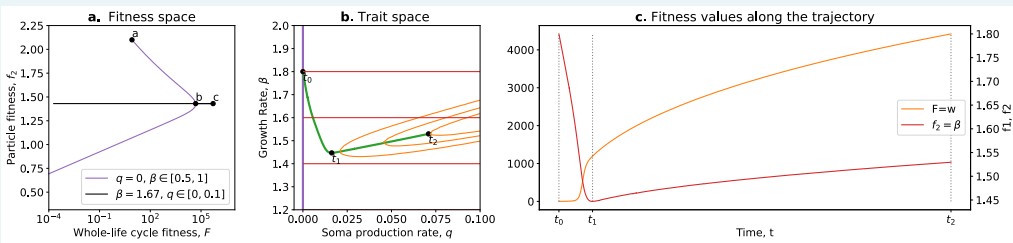

**Box 2—figure 1.** Tradeoff breaking in the ecological scaffolding scenario.
(**a**) Values of $F$ and $f_2$ accessible to the organism when $q = 0$ and only $\beta$ can mutate (purple) and values of $F$ and $f_2$ accessible to the organism when only $q$ can mutate and $\beta$ is such that $F$ is maximum for $q = 0$ (black); (**b**) trait space with isolines of fitness ($f_2$ in red, $F$ in orange), with an example of evolutionary trajectory in green (ancestral tradeoff represented in purple).
(**c**) $f_2$ and $F$ values along the example evolutionary trajectory. The times marked by vertical lines in (**c**) correspond to the dots in (**b**). Note that $F$ and $f_2$ have opposed dynamics from $t_0$ to $t_1$ (fitness-decoupling observation) and both increase from $t_1$ to $t_2$. Rare mutations on $q$ allow breaking away from the ancestral tradeoff line (tradeoff-breaking observation). Parameters: $N = 1e + 06$; $T = 30$; $d = 0$; $\rho = 0.01$; $\sigma_\beta = 0.1$; $\sigma_q = 0.001$.

where the optimal trait values are potentially different. We assume here that this change is abrupt and not accompanied by the immediate disappearance of the underlying constraints that bear upon the cell traits, particularly tradeoffs, due to the rest of their genetic machinery. Collective formation is the first phase of our scenario and is often the first phase of any descriptions of an ETI (**Bourke, 2011**; (c) **Rose and Hammerschmidt, 2021**; **van Gestel and Tarnita, 2017**). This has been studied in

## Box 3. Ratcheting and tradeoff breaking.

Tradeoff-breaking mutations are equivalent to rare mutations that change the set of accessible phenotypes. *Libby et al., 2016* propose a mechanism of ratcheting mutations that stabilises multicellularity by constraining evolutionary reversion towards unicellularity. They consider a nascent multicellular organism that switches between a multicellular $G$ and unicellular state $I$, growing in an environment alternating between two states, one favouring the multicellular life cycle ($E_G$) and the other favouring the unicellular life cycle ($E_I$). Two types of ratcheting can occur: first, mutations that improve the fitness within the multicellular type that come at a cost to the free-living type (reducing the fitness of revertants) and, second, a type of ratcheting mutation that decreases the probability that a mutation results in reversion. In the following, we show how the slowest of type 1 or type 2 ratcheting fits as a tradeoff-breaking mechanism, as presented in the main text.

The population dynamics of both types $G_t$ and $I_t$ in an environment that fluctuates between $E_G$ and $E_I$ after a fixed number of generations in each ($n_g$ and $n_i$, respectively) is given by Equation 2.3 from *Libby et al., 2016*

$$\begin{pmatrix} G_{t+ng+ni} \\ I_{tng+ni} \end{pmatrix} = \begin{pmatrix} (1-ci)(1-p)+1 & p \\ (1-ci)p & 1+(1-p) \end{pmatrix}^{n_i} \begin{pmatrix} 1+(1-p) & (1-c_g)p \\ p & (1-c_g)(1-p)+1 \end{pmatrix}^{n_s} \begin{pmatrix} G_t \\ I_t \end{pmatrix}$$

$$= \mathbf{A}_I^{ni} \mathbf{A}_G^{ng} \begin{pmatrix} G_t \\ I_t \end{pmatrix} = \mathbf{A} \begin{pmatrix} G_t \\ I_t \end{pmatrix},$$

where $p$ is the probability for cells to switch from one type to the other, and $c_g, c_i > 0$ are the fitness differences between $G$ and $I$ cells in $E_G$ and $E_I$ environmental states, respectively. In the following, we fix $c_g = 0.1$, and the traits that can mutate are $p$ and the fitness gap $\Delta c = c_g - c_i$.

We now recast this model within our framework. The whole life cycle fitness of the organism $F$ is the log of the dominant eigenvalue of the matrix $\mathbf{A}$; the counterfactual fitness $f_2$ is the fitness of the organisms if they would always be in the $E_I$ environment ($n_g = 0$)—that is, the log of the dominant eigenvalue of $\mathbf{A}_I$. Conversely, the within-collective fitness $f_1$ is the fitness of the cells if they would constantly be in the $E_G$ environment—thus, the log of the dominant eigenvalue of $\mathbf{A}_G$.

Let $(\Delta c, p)$ be the trait vector characterising a phenotype, and let us model evolution using the equation from Section 4, *Equation 3*, considering that mutations affecting $p$ (ratcheting type 2) are more frequent than mutations affecting $\Delta c$ (ratcheting type 1): $\sigma_{\Delta c} = 10^{-4}, \sigma_p = 0.2$. Initial trait values are $\Delta c = 0$ and $p = 10^{-5}$ (no fitness gap and very rare switch).

*Box 3—figures 1b and c* show the result of the simulation. The evolutionary trajectory can be split into three phases: first, a fast-paced phase where the switch probability *p* increases to 0.2, corresponding to optimisation on the tradeoff (before $t_1$, note that here *p* plays the same role as $\theta$ in Section 3, giving the linear abscissa of the cells on the tradeoff line); then a slow increase in $\Delta c$, corresponding to the slow accumulation of tradeoff-breaking mutations (type 1 ratcheting) leading to a tradeoff-breaking observation (between $t_1$ and $t_2$); and, finally, a new decrease in switch probability (type 2 ratcheting), corresponding to an optimisation on the (new) tradeoff (after $t_2$). The result is an overall increase in the proportion of G-type in the population (*Box 3—figure 1c*). Note that $F$ always increases along the trajectory. However, $f_2$ decreases in the first phase and increases in the second and third phases, showing trajectories for which the fitness-decoupling model cannot easily account.

To summarise, this simple set of hypotheses (ratcheting types 1 and 2, with mutations for ratcheting 1 being rarer) leads to a fitness-decoupling observation when selection first acts along the switch probability tradeoff (leading to higher switching and an increase in multicellular types) because the optimal trait values for $F$ and $f_2$ are different. Then, rarer type

2 ratcheting mutations result in tradeoff breaking, eventually resulting in a second (relatively fast-paced) optimisation on the switch-probability tradeoff (leading to reduced switching and entrenchment of the multicellular type), which does not result in a fitness-decoupling observation because the optimal trait values for $F$ and $f_2$ coincide. Note that, here, the tradeoff stems from the ratcheting mechanism and the environment periodically switching between multicellularity or unicell-favouring, rather than the genetic architecture (as in the main text) or population structure (as in *Box 2*).

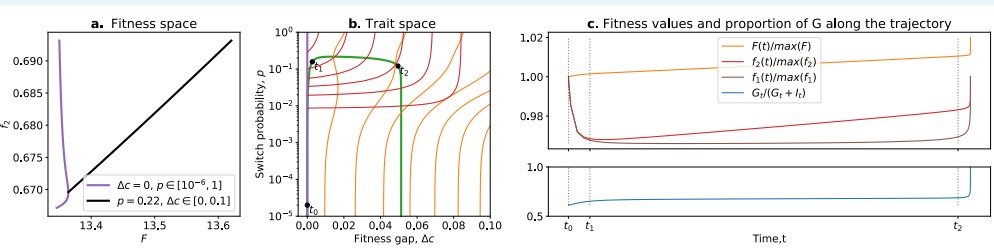

**Box 3—figure 1.** Ratcheting and tradeoff breaking.
(**a**) Values of $F$ and $f_2$ accessible to organisms when $\Delta c = 0$ and $p$ is free (purple), and when $p = 0.2$ and $\Delta c$ is free (black). (**b**) Trait space with isolines of fitness ($f_2$ in red, $F$ in orange) with an example of evolutionary trajectory in green (ancestral tradeoff represented in purple). (**c**) Fitness values for $f_1$ (in brown), $f_2$, and $F$, in addition to the stable proportion of $G$ (in blue) along the example evolutionary trajectory. The times marked by vertical lines in (**c**) correspond to the dots in (**b**). Note that $F$ and $f_2$ have opposed dynamics from $t_0$ to $t_1$ (fitness-decoupling observation), and both increase after $t_1$. Rare mutations on $\Delta c$ allow breaking away from the ancestral tradeoff line (tradeoff-breaking observation) after $t_1$. Parameters: $c_g = 0.1$; $\sigma_{\Delta c} = 10^{-4}$, $\sigma_p = 0.2$.

detail by *Tarnita et al., 2013*, particularly in the context of competition with the unicellular ancestor. In contrast, our model focuses on what happens after the initial formation of collectives.

In a second phase, we expect changes in traits to occur due to low-hanging fruit mutations, which remain constrained by the ancestral tradeoffs. We call this phase 'optimisation on the tradeoff', and it leads the system towards an optimal phenotype for the within-collective environment ('Tradeoff-optimal collective phenotype'; orange disk in *Figure 8*). We observe trait optimisation on the tradeoff in all our example model systems (*Figure 8*), except for cyanobacteria. Here, in contrast to the other empirical examples, we do not follow an ETI in progress but one that happened in the evolutionary past so that phase 2 cannot be observed (anymore) in the known multicellular species of today. During this phase in our scenario, fitness decoupling might be observed. For this to occur, the system must meet a few assumptions we made in the previous section: namely, that fitness is a continuous function of traits, and that the ancestral traits are initially optimal for the single-cell environment—and, thus, optimal for counterfactual fitness—but not optimal for the within-collective environment. When these assumptions are met and selection drives the system towards better trait values (within constraints) for this new within-collective environment, counterfactual fitness will necessarily decrease while collective fitness will necessarily increase (from the blue to the orange disk in *Figure 8*). If this set of assumptions is violated, nothing can be said about the relationship between counterfactual and collective fitness.

The third phase entails changes in phenotype that lie outside the constraints of the ancestral particles. These are driven by selection of the system towards new, previously unreachable trait values ('unconstrained optimal collective'; red disk in *Figure 8*). We refer to this third phase as the tradeoff-breaking phase (see *Figure 8* for adaptations in our example model systems). During this phase, particle fitness—whether counterfactual ($f_2$) or within collectives ($f_1$)—does not necessarily continue to decrease, even if the conditions for fitness-decoupling observations outlined earlier are fulfilled. This is so because the new trait values are in the region of the trait space that was not reachable by the ancestor. In consequence, there is no particular theoretical or biological reason to expect that

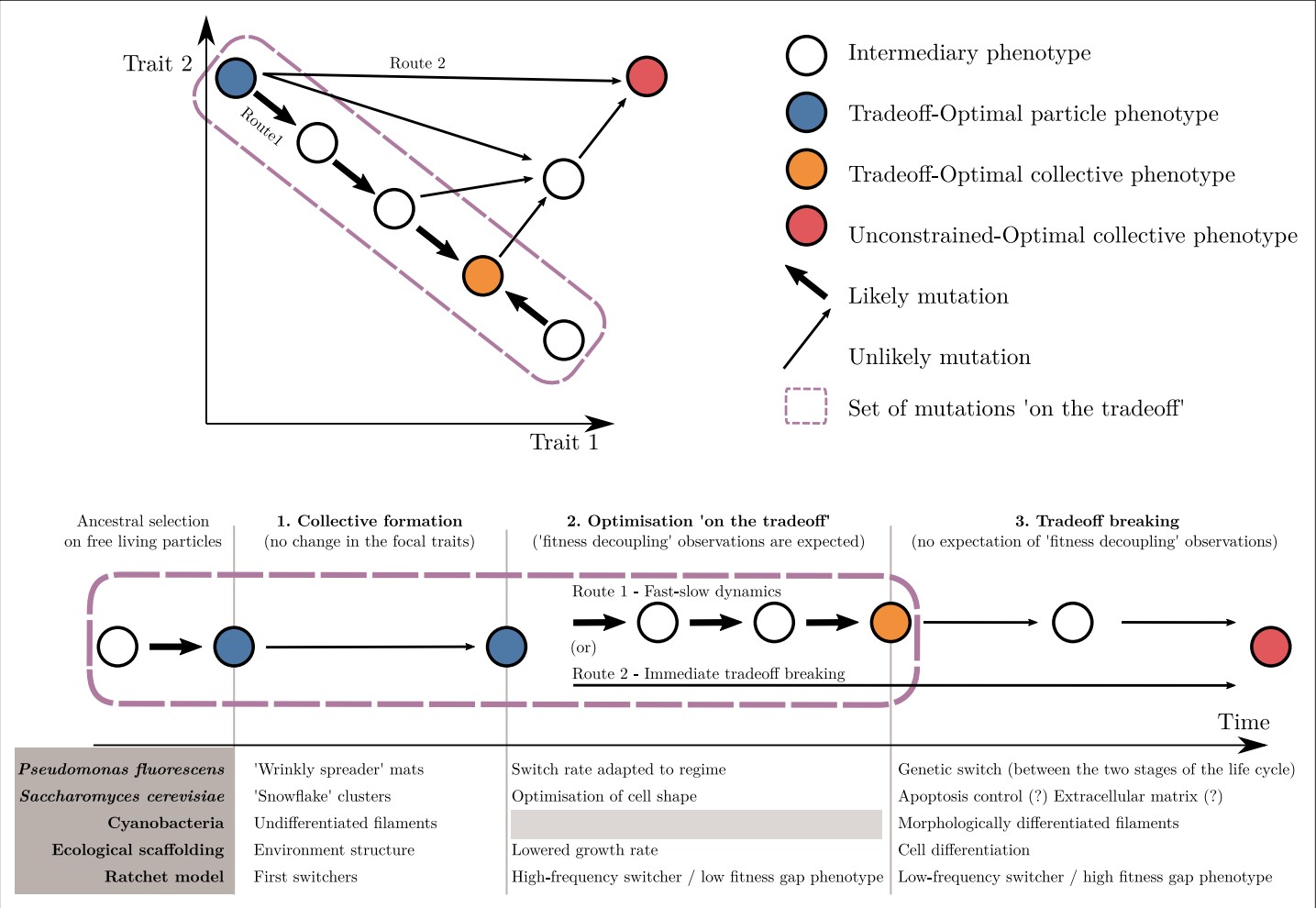

**Figure 8.** An adaptive scenario for evolutionary transitions in individuality as a consequence of the trait-based approach. (1) Collective formation of particles occurs through an event that does not change the focal traits. (2) Optimisation 'on the tradeoff', where the traits are selected within the constraints passed down from the ancestral organism. (3) Tradeoff breaking, where mutations that are not bound by the ancestral constraints enable tradeoff breaking. Fitness-decoupling observations are expected in the second phase but not in the third. Note that the second phase might be shortened or even bypassed depending on the mutational path (this second route 'immediate tradeoff breaking' contrasts with the previously described first route 'fast-slow dynamics'). This framework can be used to interpret a diversity of experimental and mathematical model systems as shown with the examples: *P. fluorescens* from **Hammerschmidt et al., 2014**, *S. cerevisiae* from **Ratcliff et al., 2015**, ecological scaffolding from **Black et al., 2020**, and ratchet model from **Libby et al., 2016**.

they would have been suboptimal and selected against, had the ancestor been endowed with such trait values.

If we assume that tradeoff-breaking changes are less likely than the low-hanging fruit mutations constrained by a tradeoff, we statistically expect them to occur predominantly after the second phase, resulting in a 'fast-slow dynamic' (route 1 in **Figure 8**). This route is the focus of this article. However, while we do not develop this in detail here, depending on the biological system, the second phase might be short if tradeoff-breaking changes are the first to occur. This possibility is particularly relevant for experimental studies. A short second phase implies that it could easily be missed by experimenters. Another possibility is that the second phase is missing entirely (route 2 in **Figure 8**).

Yet another possibility not studied in this article is that tradeoff-breaking-like observations could be made immediately after or even during the formation of a collective, leading to the spontaneous emergence of properties that also confer the collective living benefits. Thus, tradeoff-breaking dynamics (whether following route 1 or route 2 in **Figure 8**) could be driven entirely by the phenotypic plasticity of particles, without the need for mutation. For instance, in clonal collectives, one could imagine that phenotypic heterogeneity leads to a primordial form of division of labour (**Ispolatov et al., 2012**;

*Ulrich et al., 2018*; *van Gestel and Tarnita, 2017* for details), which would provide such collectives with an immediate selective advantage (*Ispolatov et al., 2012*; *Ulrich et al., 2018*). In this 'plasticity first' scenario, the different phenotypic states would already be part of the genetic repertoire of the unicellular ancestor.

Our main objective in advocating this trait-based approach is to offer a consistent framework for interpreting models, empirical observation, and evolutionary experiments of ETIs, which can accommodate the apparent fitness-decoupling observations, in addition to the tradeoff-breaking ones, without referring to the export-of-fitness metaphor. Provided this empirical objective, it is legitimate to ask to what extent the tradeoff-breaking scenario is representative of ETIs and can be used as a marker that an ETI is underway or has occurred. First, one must recognise that tradeoff breaking might occur outside ETIs. For instance, tradeoff breaking and, more generally, 'constraints breaking' are already considered key events in the evolution of body plans and are expected to be a widespread mechanism for the emergence of novelties (see *Galis and Metz, 2007*; *de Vos et al., 2015*). Thus, a tradeoff-breaking event *in and of itself* cannot be used as a marker of an ETI. However, considering a trait in the context of an ETI, where collectives are formed and maintained (phases 1 and 2), and establishing how those collectives change and potentially overcome the constraints that historically bear on particles (phase 3) offers a good empirical handle to study ETIs. Conversely, the lack of any tradeoff-breaking observation in such an empirical system could be used as an indication that the transition is still in its early stages.

## Discussion

Fitness-centred approaches to ETIs have been influenced by the concept of fitness-decoupling between lower-level particles and higher-level collectives. In this view, the fitnesses of particles and collectives are initially proportional to one another but diverge as an ETI occurs—particle fitness decreases while collective fitness increases. This interpretation comes with some inconveniences. First, fitness is notoriously difficult to define and measure. This, in turn, makes fitness comparisons across levels difficult. Second, fitness values in and of themselves do not provide a mechanistic model of the system. Progress in understanding ETIs relies on our ability to circumvent limitations inherent in the currency of fitness. We suggest that, to study ETIs, focusing on traits and tradeoffs between traits, rather than focusing on fitness, is both more parsimonious and practically achievable. Finally, we propose that rare tradeoff-breaking events are a crucial part of ETIs and could be used experimentally for their detection.

Our first main finding is a new formal argument cementing the position that decoupling between commensurable measures of fitness is impossible. Starting from the recognition that fitness is a concept difficult to define consistently (*Abrams, 2012*; *Ariew and Lewontin, 2004*; *Doulcier et al., 2021*), the problem is magnified when the entities to be compared belong to different levels of organisation. As we discuss in Section 1 (and *Box 1*), experimentally comparing fitness in such cases would require being able to measure the growth rates in the same environment at different levels of organisation, which proves challenging. Even if such a comparison could be made, fitness measures at different levels of the same biological substrate necessarily lead to the same outcome at any level. We show this point formally in Section 2. In particular, commensurability is assured by taking care to use the same set of events (same reference environment and same timescales) for both measures. Once this is ensured, fitness decoupling is not observed. Thus, our analysis reveals that fitness-decoupling observations result from incommensurable fitness measurements. We formally confirm the analysis provided by *Bourrat, 2015a*; *Bourrat, 2015b* (see also *Black et al., 2020*), who qualify such observations as artefacts of descriptions and confirm the result of *Shelton and Michod, 2014* that 'cells and colony level fitness […] are mathematically equivalent ways of bookkeeping' (p. 457) by extending it to more general models of population dynamics. Counterfactual particle fitness does not suffer from this problem and can vary independently of actual fitness. Thus, it can be used to characterise evolutionary tradeoffs in the context of ETIs. However, we highlight the fact that this counterfactual value is not compatible with the fitness transfer metaphor because it is a theoretical construct and not an actual quantity to be transferred (but see *Shelton and Michod, 2020*, for a diverging opinion on the subject).

Our second main result is a general adaptive scenario of ETIs that is compatible with both experimental fitness-decoupling observations (artifactual or counterfactual) and tradeoff-breaking deviations.

In Section 3, we clarify the conditions under which a fitness-decoupling observation (between incommensurable fitness measures) could, in principle, be made—using a simple tradeoff model between trait values, we find that one condition is that the optimal trait values for counterfactual particle fitness and whole life cycle fitness are different. In Sections 4 and 5, we show that if the tradeoff is relaxed through the existence of rare tradeoff-breaking events, fitness-decoupling observations may not hold for whole evolutionary trajectories. We suggest that an evolutionary trajectory can be divided into three phases: a first phase during which collectives are formed, followed by an optional second fast-paced phase during which optimisation 'on the tradeoff' occurs, and, finally, a slower phase driven by rare tradeoff breakings. Tradeoff-breaking mutations might result in lineages where both counterfactual and whole life cycle fitnesses are increased compared to the ancestor. We propose that departures in collective-level entities from ancestral tradeoffs—tradeoff-breaking points—are a mark of a key moment in ETIs and might be used to characterise them. This proposal is compatible with recent models found in the literature on ETIs—namely, the ecological scaffolding model (*Black et al., 2020*; *Bourrat, 2022*; *Doulcier et al., 2020*) and the ratchet model (*Libby et al., 2016*; *Libby and Ratcliff, 2014*)—that provide alternative mechanisms for both tradeoffs and tradeoff-breaking observations. Further, we show that it is also compatible with experimental data on cyanobacteria (*Colón-López et al., 1997*; *Misra and Tuli, 2000*; *Berman-Frank et al., 2001*; *Berman-Frank et al., 2003*; *Popa et al., 2007*; *Mohr et al., 2013*) and *P. fluorescens* (*Hammerschmidt et al., 2014*; *Rose et al., 2020*).

Our tradeoff-breaking framework could also serve as a springboard to generate new hypotheses. The study of tradeoff breaking requires estimating changes between the pre-ETI ancestral traits (e.g., unicellular) and post-ETI derived traits. Access to ancestral traits can be gained in multiple ways, depending on the system studied. The first is through phylogenetic reasoning, such as by reconstructing the sequence of ecological and phenotypic trait evolution during the evolution of cyanobacterial multicellularity (as in *Hammerschmidt et al., 2021*). The second is by assuming that the ancestral traits are close to the optimal values with respect to the counterfactual particle 'outside of the collective'. The counterfactual method described by *Shelton and Michod, 2014* is useful here. The third is through direct measurement during experimental evolution studies (*Ratcliff et al., 2012*; *Hammerschmidt et al., 2014*; *Herron et al., 2019*). Additionally, statistical methods to better characterise tradeoff breaking should be developed.

Fitness-centred approaches to ETIs may have reached their limits. We propose to refocus the problem on tradeoffs between traits, thereby bypassing the difficulties inherent to fitness comparisons. The advantages of this move are multiple and range from allowing better experimental accessibility to producing a more mechanistic theory. The way the collective-level context affects the constraints that link traits together is the linchpin of our framework. In particular, we argue that tradeoff-breaking events represent a mark of significant evolutionary innovation towards individuality at the higher level that might be missed by fitness-centred approaches.

## Acknowledgements

We thank María Rebolleda-Gómez, Matthew Herron, Corina Tarnita, and Will Ratcliff for review of the manuscript and valuable comments. We also thank the Theory & Method in Biosciences group at the University of Sydney and Macquarie University, for their feedback on a previous version of the manuscript. Finally, we thank Sarah Pearce for editing the mansuscript.

## Additional information

### Competing interests
Paul B Rainey: Reviewing editor, eLife. The other authors declare that no competing interests exist.

### Funding

| Funder | Grant reference number | Author |
|---|---|---|
| John Templeton Foundation | 62220 | Pierrick Bourrat Katrin Hammerschmidt Guilhem Doulcier |

| Funder | Grant reference number | Author |
|---|---|---|
| Max Planck Institute for Evolutionary Biology | Open access funding | Paul B Rainey Guilhem Doulcier |
| Australian Research Council | DE210100303 | Pierrick Bourrat |

The funders had no role in study design, data collection and interpretation, or the decision to submit the work for publication.

## Author contributions
Pierrick Bourrat, Conceptualization, Funding acquisition, Methodology, Writing – original draft, Writing – review and editing; Guilhem Doulcier, Conceptualization, Formal analysis, Software, Visualization, Writing – original draft, Writing – review and editing; Caroline J Rose, Paul B Rainey, Conceptualization, Writing – review and editing; Katrin Hammerschmidt, Conceptualization, Data curation, Writing – original draft, Writing – review and editing, Funding acquisition

## Author ORCIDs
Pierrick Bourrat http://orcid.org/0000-0002-4465-6015
Guilhem Doulcier http://orcid.org/0000-0003-3720-9089
Paul B Rainey http://orcid.org/0000-0003-0879-5795
Katrin Hammerschmidt http://orcid.org/0000-0003-0172-8995

## Decision letter and Author response
Decision letter https://doi.org/10.7554/eLife.73715.sa1
Author response https://doi.org/10.7554/eLife.73715.sa2

# Additional files

## Supplementary files
• Transparent reporting form

## Data availability
The code implementing the models is publicly available on Zenodo (https://doi.org/10.5281/zenodo.5352208). For Figure 1: Protocol described and statistical analysis performed in Hammerschmidt et al. (2014). Dataset published as Rose et al. (2018). For Figure 6b: Data taken from Colon-Lopez et al. (1997); Mohr et al. (2013); Misra & Tuli (2000); Berman-Frank et al. (2001); Popa et al. (2007) and standardised. For Figure 6c: Data taken from the dataset published as Rose et al. (2018).

The following previously published dataset was used:

| Author(s) | Year | Dataset title | Dataset URL | Database and Identifier |
|---|---|---|---|---|
| Rose CJ, Hammerschmidt K, Pichugin Y, Rainey PB | 2018 | Meta-population structure and the evolutionary transition to multicellularity | https://zenodo.org/record/3748416#.YwTOaOzML0q | Zenodo, 3748416#.YxXSi-zML0r |

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

# Appendix 1

## Glossary

- *Particles* or *cells:* The lower-level entities of a two-level biological system.
- *Collectives* or *multicells:* The higher-level entities of a two-level biological system.
- *Evolutionary transition in individuality* (ETI): Evolutionary process during which collective-level entities become evolutionary individuals and are able to participate in the process of evolution by natural selection 'in their own right'.
- *Fitness*: The expected average exponential growth rate of a given type of individual (i.e., individual sharing the same traits) in a given steady-state reference environment.
- *Fitness-decoupling observation*: The observation that the fitness of particles decreases while the fitness of collectives increases. Used in the literature as a probable hallmark of an ETI.
- *Within-collective particle fitness:* The fitness of a particle within the collective environment, ignoring collective-level events (e.g., because they happen over a longer timescale). Noted in the main text.
- *Counterfactual particle fitness:* The fitness of a hypothetical particle with the same traits as the actual particle that would live in a non-collective reference environment. There is no unique way to define the counterfactual reference environment. Noted in the main text.
- *Whole life cycle particle fitness:* The fitness of a particle computed over a reference environment that includes the whole life cycle of collectives (including collective birth–death events). Mathematically equal to the collective-level fitness if the collective stage distribution reaches a steady state (i.e., collectives do not keep getting bigger or smaller) as proven in Section 2. Noted in the main text.
- *Export-of-fitness model:* A model used to explain *fitness-decoupling observations* by a 'transfer of fitness' from the particle level to the collective level during ETIs. We treat the terms 'fitness transfer' and 'fitness decoupling' (when referring to the interpretation not the observation) as equivalent to 'export of fitness'.
- *Tradeoff model*: An alternative model to the *export-of-fitness model* used to explain *fitness-decoupling observations* by invoking ecological or genetic constraints on the values of traits that contribute to *counterfactual* and *whole life cycle fitness* during an ETI.
- *Tradeoff-breaking observation:* The observation that some lineages do not seem to conform to the *fitness-decoupling observation* during an ETI—they show an increase of both counterfactual or within-collective fitness and collective fitness.
- *Tradeoff-breaking model:* A model where the evolutionary trajectories follow constraints that come from the unicellular ancestors (tradeoff) and that include rare phenotypic changes that are not submitted to the same constraints (tradeoff breaking). This model can account for both fitness-decoupling and tradeoff-breaking observations during ETIs.

