## [Editor Report]

This article makes two important, independent contributions to the multicellularity/major transitions literature. First, it sheds light on the concept of ‘fitness decoupling’, providing a strong mathematical foundation for the claim that the fitnesses of cells and groups cannot be formally decoupled during an evolutionary transition in individuality. Second, the article proposes using the evolution of tradeoff-breaking traits as an indication that an evolutionary transition has occurred. This dovetails well with existing fitness-based approaches, extending the toolkit of those studying major evolutionary transitions.

---

## [Decision Letter]

**Decision letter after peer review:**

Thank you for submitting your article "Beyond Fitness Decoupling: Tradeoff-breaking during Evolutionary Transitions in Individuality" for consideration by *eLife*. Your article has been reviewed by 3 peer reviewers, and the evaluation has been overseen by a Reviewing Editor and Patricia Wittkopp as the Senior Editor. The following individuals involved in review of your submission have agreed to reveal their identity: María Rebolleda-Gómez (Reviewer #1); Matthew Herron (Reviewer #2); Corina E Tarnita (Reviewer #3).

**Essential revisions:**

Fitness decoupling is an important topic that has created a lot of confusion in the field over the last decade and a half. Work clarifying this topic, and moving beyond looking at fitnesses, is welcome and important. The authors are well-poised to write this paper, bringing together real strength in the philosophy of biology, mathematical modeling, and experimental evolution of multicellularity. The modeling framework used here is useful, showing that cell and group-level fitnesses cannot be decoupled when they are both assessed over the same time scale. I agree with referee 3 that it is great to see the authors wrestling with real world data, and not relying only on simple mathematical models, because a key piece of the utility of this kind of theory will rely on people being to apply it to empirical systems. I also agree with the authors that traits, not just fitnesses, can be useful in determining when an ETI has occurred. Moving into traits, the authors propose to focus on trade-off breaking as an indicator that an ETI has occurred, which is creative and potentially quite useful, though further clarification will be required.

The referees raised several important points that will need to be addressed in the revision.

**1)** As Referee 1 points out, the idea that cell and group-level fitnesses cannot be formally decoupled was noted by Michod in his 2005, 2014, and 2020 papers. While I do not think that Michod has been entirely clear on this, his 2020 paper makes this point unambiguously. I think there's room for grace here with respect to what Michod thinks, and I'd like to avoid un-necessary strawman arguments. I don't think we should conflate Michod's views on this with Okasha's, and should give credit to Michod for accepting that cell and group-level fitness are inexorably correlated. But perhaps more importantly, the conclusion that in simple mathematical models cell and group-level fitnessess cannot be decoupled is not itself novel: Dr. Bourrat himself published this result as the central message of his 2015 paper on fitness decoupling (Bourrat, "Levels, time and fitness in evolutionary transitions in individuality". Philosophy and Theory in Biology). So I'd recommend modifying the language in the manuscript here around whether this is a novel finding (i.e., 2nd lead sentence of the discussion: "Our first main finding is the impossibility of decoupling between commensurable measures of fitness".).

**2)** I think the main contribution of this paper is the shift from focusing on fitnesses to traits. Specifically, the authors advocate for the concept of trade-off breaking as an indication that an ETI has occurred. I like this idea, in part because it is far easier to measure traits than fitnesses in real organisms. However, the referees have raised some important issues with this idea that should be addressed during revision. As Referees 1 and 2 pointed out, organisms evolve trade-off breaking traits all the time without affecting biological individuality. I thought Referee 1's example of the zebra herd foraging trait was quite effective here, illustrating how collective-behaviors that break trade-offs may be not meaningfully affect the level of biological individuality.

Most importantly, I'd like to see this idea developed a bit more to describe the conditions under which trade-off breaking is an indication of an ETI. Specifically, I'd like the authors to clarify: (1) Is every case of trade-off breaking a case of an ETI? (2) If not, are all ETIs characterized by trade-off breaking? Or (3) Is trade-off breaking one possible way of driving an ETI, but not a universal trait of ETIs? (4) If the latter is true, and trade-off breaking is not necessary or sufficient to distinguish an ETI, then what else is needed to distinguish an ETI? The authors have made a compelling case that the fitness decoupling literature has significant problems, but it is not yet clear that trade-off breaking is the solution.

To clarify my thinking on this a bit more: trade-off breaking seems like a useful tool when there are clear trade-offs that can be ameliorated by multicellular adaptation (for example, by cellular differentiation- the cyanobacterial example is a good one). But what about a case of simple multicellular adaptation in the absence of pre-existing trade-offs? For example, consider a simple organism like a snowflake yeast that is obligately multicellular, reproducing by group fragmentation. Over generations, cells evolve novel traits that improve the fitness of cells living in multicellular groups, but these traits reduce the growth rate of cells. It's easy apply the counter-factual approach and see how the same traits increase the fitness of groups and the cells living in these groups, while decreasing the counterfactual fitness of solitary cells (they grow slower). To me, this is a straightforward use of the counterfactual fitness approach, in line with what Michod suggests in their 2020 paper, and is indicative of an ETI in progress. This isn't hypothetical, it's precisely what we see in snowflake yeast (see Bozdag et al., 2021 Nature Comms for some published data, but we have much more work done in this area that is unpublished). I don't think our model system is very unusual here, and I would think that this kind of simple group-level adaptation is common during the transition to multicellularity.

Thus, the 3 referees and I agree that it is essential that the authors clarify whether trade-off breaking is a universal trait of ETIs, or is one potential diagnostic trait that an ETI has occurred. If it is not universal (which is what we suspect at this point, and claims of universality would require new, compelling arguments), then it would be useful to outline the conditions under which one might want to use a trait-based argument (i.e., trade-off breaking) vs. a fitness-based argument (i.e., counterfactual fitness). The referees and I generally thought that trade-off breaking could be a useful tool for determining that an ETI has occurred, even if it is not a diagnostic feature of all ETIs.

**3)** Referee 3 points out that the model developed in this paper is not itself novel, being similar in structure and analysis to her prior work. Please cite/discuss this prior work appropriately.

*Reviewer #2 (Recommendations for the authors):*

Figure 2 would be more useful if it included a comparison of fitnesses, for example between different genotypes (in other words, if it compared particle- and collective-level fitnesses of two genotypes and showed that they are identical if measured over the same time span).

*Reviewer #3 (Recommendations for the authors):*

Let me start by saying the I am sympathetic to the authors' position on fitness decoupling, which I've always found to be so contrived as to be either non-sensical or meaningless, depending on how one defines fitness. I also quite like the simple proposals of sections 3 and 4 and appreciate that the authors engaged both with empirical evidence and with ways to see existing theoretical models through the lens of the tradeoff-breaking framework. This being said I have a few concerns that I would like addressed:

**1.** the writing is fairly dense and convoluted: it took me several hours to go through the manuscript, and I'm still unsure that I'm fully grasping all details. If this is going to be engaged with by a broad audience, and especially a non-specialist one, the authors need to make more of an effort to clean up, streamline, and provide intuition. For instance,

– I can't make sense of Figure 4 I don't know which ones of the dots are particles; I don't know which is the focal particle whose fitness we are tracing, I don't understand what the vertical dotted lines are, and I don't know why there are multiple horizontal lines, seemingly associated with both blue and orange dots. This lack of clarity around exactly how to measure f2 and F propagate, making it sort of hard to rigorously understand what the authors meant f2 to be in various of the examples (though, in principle, if I don't worry too much about understanding Figure 4, I can follow what f2 in principle might be in each example, as needed).

– I found it tedious to go through all the math derivations and references to theorems in the main text, with, often, little to nothing by way of biological intuition that would help ground some of those results. When some biological intuition is provided, it often left me wanting some more explanation. For instance, in discussing the example of section 3.3, the authors state: "Note importantly that the opposite directions of the dynamics of F and f2 is a consequence not a cause of the underlying tradeoff mechanism that drives the evolutionary dynamics." It would be great if the authors could explain why this is; and, in general, this whole example could use a bit of biology on the very bare theoretical bones. For example, it seems that all collectives of size i >= 2 have the same survival probability and the same probability to grow to the next class size; if the authors can't contextualize this with a bit more biology, at the very least it would be nice to have a sentence that simply states this assumption in words. A non-mathematical reader might have a hard time inferring that from the fact that the expressions of p_i and g_i don't depend on i.

– Although I understand the tradeoff in Black et al., I'm still not entirely sure I understand the tradeoff in the model in Box 2: is there supposed to be a tradeoff linking β and q? It seems that β is a growth rate whose increase would be good for both germ cells and soma cells. In both Boxes 2 and 3 it would be helpful to have it more clearly spelled out which modeling aspects are new to this paper and which ones are taken from the original papers, which were themselves modeling papers.

– Simultaneously, some things that are simple to state intuitively, are presented in a somewhat convoluted and tedious way: e.g., the 1st paragraph of Box 1 and associated Figure B1a make overly complicated the story that any reader would appreciate with a simple example: a bacterium that doubles every hour will make more of itself in one year than a mouse that "doubles" every year (or interest compounding or a suite of other simple, intuitive examples). I also think that some such intuitive discussions should come earlier, as the authors set up the discussion of fitness in the main text (prior to Figure 4).

– In general, I'm not convinced that some of the mathematical detail needs to be in the main text as opposed to a detailed methods section.

**2.** The model in Section 2 is not new. Tarnita et al. 2013 J. Theor Biol (Evolutionary construction by staying together and coming together) have used the same model to explore the origin of multicellularity. Even though that paper looks at competition of such collectives against the solitary ancestor, the ideas, setup and analyses are nearly identical to what the authors do in section 2.

**3.** This brings me to my final and broadest points:

– First, if I understand correctly, in this manuscript Bourrat et al. do not seem to consider competition against an extant unicellular ancestor anymore. But, it seems to me that one of the greatest challenges would've been the persistence of the multicellular lifestyle in the earliest stages, before rare mutations can break tradeoffs and whatnot. It would be useful to at least have some discussion of this early on, so that I can situate where exactly the modeled worlds are in the origin of multicellularity and transition to individuality (i.e. what is assumed to have happened already, vs what is the new challenge to be explained).

– Second, how should we think of free riders in this framework? They don't seem to be explicitly included in any of the modeling.

– Third, much as in the original Michod thinking, it seems to me that the current framework also sort of assumes that the collective is a sum of what the component cells do and does not consider spontaneous emergent properties that simultaneously confer benefits of group living on the group and also on the parts, such as some degree of division of labor [that harnesses the repertoire of the unicellular ancestor and arises e.g., due to emergent information gradients in the earliest collectives]. Such spontaneous benefits could in themselves set the stage for an ETI without requiring fundamental reversals of tradeoffs (even though these of course could follow later). Again, I'm not suggesting that the current framework should be explicitly modified to account for such dynamics, but it would be informative to have at least some discussion of whether the authors believe that tradeoff-breaking by rare mutations is the only way to get an ETI or whether it's one possible thing to look for empirically, but not necessarily the only route.

To contextualize this point further, it might be helpful to look at Ulrich et al. 2018 Nature (Fitness benefits and emergent division of labour at the onset of group living) and Brahma et al. 2017 (Emergence of cooperation and division of labor in the primitively eusocial wasp Ropalidia marginata). Both of these look at the evolution of eusociality rather than multicellularity, but the essence of the ETI can be captured by the same model (see, Tarnita et al. 2013). I'm not aware of similar empirical work in multicellularity, though see Gavrilets 2010 PLoS Comp Biol (Rapid Transition towards the Division of Labor via Evolution of Developmental Plasticity) for theoretical work.

*[Editors' note: further revisions were suggested prior to acceptance, as described below.]*

Thank you for resubmitting your work entitled "How Tradeoff-Breaking Events Go Beyond Fitness Decoupling During Evolutionary Transitions in Individuality" for further consideration by *eLife*. Your revised article has been evaluated by Patricia Wittkopp (Senior Editor) and a Reviewing Editor.

The three referees and I have read your revised manuscript, and really appreciate some of the new additions. In particular, there is a lot of enthusiasm for the new Section 5. Everyone found the paper to be thought-provoking and thought that this paper will make an important contribution to understanding ETIs if suitably revised.

The main thing we would like to suggest is a more inclusive tone for the paper. I do not think it takes away any of the novelty of the results, and indeed, can strengthen the scholarship of the work. It is critical that you revise the introduction to more fairly reflect Michod et al.'s stated meaning (not the authors' inferred meaning) about the concepts of export of fitness, fitness decoupling, and the counterfactual account. Second, the referees and I do not see why the counterfactual account is not a valid indicator that an ETI has occurred. This does not undercut the utility of your work, showing that trade-off breaking mutations can be an indication of an ETI. But all four of us are convinced that both approaches have merit.

Referee 2 has left a detailed review, highlighting five major points. The referees and I agree that it is essential to address all five points with changes to the MS, not arguments for why revision is not necessary. Please respond to their review thoroughly. I have carried forward their five summary points and provided a bit of additional context on some below, reflecting our collective post-review discussion.

**1.** Adequately address Michod and colleagues' "counterfactual fitness" interpretation of their own model or give a compelling reason not to in the manuscript.

Despite pointing this out in the last revision, the paper still does not treat Michod et al.'s position on fitness decoupling and counterfactual fitness fairly. The paper is framed in such a way that it critiques a straw-man version of Michod et al's position on these topics (privileging the authors' interpretation of what Michod et al. have said, or what the metaphor of "fitness decoupling" implies), which stands in contrast to what Michod and colleagues have explicitly said they mean by these terms in their more recent papers.

**2.** Make a compelling case for the lack of utility of the counterfactual fitness version of fitness decoupling in the manuscript or remove this claim from the manuscript.

Given that you are using a Darwinian Individuality approach to define what is an individual (clearly defined in the glossary, thank you!), I do not see how one can dismiss the utility of the counterfactual fitness approach. Empirical measures consistent with the counterfactual fitness account can provide insight into whether groups have been acting as Darwinian Individuals. For example, consider a lineage evolving along a pre-existing trade-off between traits that improve the fitness of groups and those that improve the fitness of cells, like those identified in Figure 8's depiction of 'optimization on the trade-off'. Using the example given in this section, consider the mutations that make snowflake yeast cells elongate, which increase group size (and thus fitness) but decrease cell growth rates (and thus fitness). This would be detected in the counterfactual fitness account and would rightly be interpreted as a trait that evolved due to selection acting among groups, not among individual cells, demonstrating that groups are Darwinian Individuals. Trade-offs have not been broken, but nonetheless, groups replicate, have heritable variation in group-level traits, and selection acts on those traits, resulting in adaptations that would not have occurred among unicellular populations. I think we can all agree that in this case, measurements consistent with the counterfactual approach would indicate that groups are a Darwinian Individual according to PGS's criteria, right? On a related note, I couldn't figure out why the MS embraced the 'Ratcheting' model but rejected the counterfactual account since ratcheting is based on the same trade-offs that would be detected by the counterfactual fitness account.

However, I think your paper raises very valid limitations on the scope of counterfactual fitness- not all multicellular adaptations will necessarily create a trade-off between fitness in groups and fitness in isolated cells, reverting lineages to unicellularity may often prove impossible, and measuring fitness can be difficult, etc. From my perspective, both the counterfactual fitness account and the trade-off breaking account may be useful indicators that an ETI (following the Darwinian Individuality criteria) has occurred. And importantly, I do not think either is necessary or sufficient to show an ETI has occurred without additional information about the traits themselves. They both require context to be informative.

Since neither is necessary for an ETI, a lineage may evolve along either route (or both simultaneously, as they are not mutually exclusive). I could imagine cases where it is much easier to identify trade-off breaking mutations and others where it is easier to identify counterfactual fitness. Having both tools available is a net positive for our field, and I don't see the need to limit ourselves only to tradeoff-breaking mutations. All the more so if trade-off breaking mutations are not necessary or sufficient to show an ETI, making the decision to only consider them somewhat arbitrary.

**3.** Consistently describe what is and is not novel about the finding of incommensurability.

This is described clearly in the original review and I have no additional context to add.

**4.** Consistently acknowledge that trade-off breaking is insufficient as a marker for an ETI.

As acknowledged in the prior review, trade-off breaking is not sufficient to determine that an ETI has occurred. It is possible to have a trade-off-breaking mutation occur within a single level. The referees and I agree that the MS should be consistent about this limitation and avoid saying the existence of that trade-off breaking mutations mean an ETI has occurred. To this end, it would be very helpful if the authors could identify what additional information one would need to know in order to conclude that a trade-off breaking trait indicates that an ETI has occurred (i.e., it's pretty clear if the adaptation derives from cellular differentiation in a way that overcomes cell-level trade-offs, like C and N fixation in cyanobacteria, but what about other kinds of adaptations? How would one infer that adaptations generating novel multicellular morphology are breaking trade-offs? What if the adaptations in question do not stem from the limitations of the unicellular ancestor [i.e., those that are limited by trade-offs], but rather reflect the novel ecology and behavior of the multicellular organism?).

**5.** Clarify, in the manuscript, how the dynamics of F and f2 fail to explain the dynamics described in lines 407-417.

This is described clearly in the original review and I have no additional context to add.

*Reviewer #1 (Recommendations for the authors):*

This paper will be an important read for everyone interested in discussions about fitness, transitions in individuality, and innovations more broadly. It helpfully pushes the conversation away from fitness and into understanding the underlying traits.

This revised version frames the argument better within its context in the previous literature, and it does a good job of clarifying central points that were hard to follow in the previous version. Figure 8, the new discussion of the snowflake yeast example, and the added text at the end of the trade-off breaking section add clarity and set the boundaries of what is and what isn't explained by this framework.

*Reviewer #2 (Recommendations for the authors):*

The revised manuscript largely fails to adequately address the central criticisms in my review of the original manuscript. Reasons for this are given in the authors' response, but these responses are inadequate for the reasons detailed below.

1. In my review of the original manuscript, I suggested that the manuscript did not fully address Michod and colleagues' description of their fitness decoupling models in terms of "counterfactual" fitness ("the impression conveyed is that Michod and colleagues consider decoupling of "commensurably computed fitnesses" possible, which is counter to their explicit statements on the topic."). The revised version still does not. Counterfactual fitness is not introduced at all until Section 2.3, so the Introduction and much of the Results read as if Michod and colleagues had never clarified that this is how their fitness decoupling models should be understood, and the authors' preferred interpretation, was explicitly rejected by Michod and colleagues, is central to the manuscript's framing. The authors' response gives some (inadequate) rationale for largely ignoring Michod and colleagues' preferred interpretation of their own model, but this rationale is not included in the manuscript. The reason I say the rationale is inadequate is that it privileges the authors' interpretation (what the "metaphor implies") over Michod and colleagues' explicit, repeated, published explanations of their own model. If the authors wish to frame the manuscript in terms of a version of Michod and colleagues' model that Michod and colleagues themselves say is mistaken, an adequate justification for doing so should be included in the manuscript.

2. In my review of the original manuscript, I said that "the final assessment (Section 2.3) does not make a compelling case for its central argument, the lack of utility of the fitness decoupling concept." The relevant section (actually 2.4) still does not make a compelling case that the fitness decoupling concept lacks utility. The reason given is "because they [F and f1/f2] are not commensurable." Since commensurability of F and f2 is not part of Michod and colleagues' formulation of fitness decoupling, this is not a compelling reason. As I said in my original review, "…'because they are not commensurable' does little to explain why the counterfactual interpretation of fitness decoupling 'does little on its own to clarify the process of an ETI,' since commensurability is not a claim that the counterfactual interpretation of fitness decoupling makes." The authors' response is inadequate for reasons I have touched on before (privileging the authors' interpretation over Michod and colleagues' explicit description of their own model), and the manuscript has not been modified to address this point.

3. The Review Editor points out that "the conclusion that in simple mathematical models cell and group-level fitness cannot be decoupled is not itself novel", and the authors' response acknowledges "this idea has already been developed in the literature"; however:

– The Abstract reads (lines 34-37) "Using a mathematical approach, we show this concept to be problematic in that the fitness of particles and collectives can never decouple-calculations of particle and collective fitness performed over appropriate and equivalent time intervals are necessarily the same." The Abstract reads as if this were a novel finding of this work, which it is not.

– Similarly, lines 95-97 and 109-110 suggest the novelty of this result.

– Lines 190-191 read "While fitness decoupling and the export-of-fitness model might seem useful concepts for understanding ETIs, we show formally here that commensurably computed fitnesses cannot generally be decoupled." Again, this sounds like a novel result, which it is not. The subsequent sentences provide some context but do not acknowledge that Michod derived this result in 2005. It may be that the authors intend some distinction from prior results as suggested in the authors' response (e.g. that they were not shown "formally" or "generally"), but this distinction is not clear in the manuscript. What is novel about this result and how it differs from prior results (including the formal analysis of Shelton & Michod 2014) will need to be clarified in the manuscript.

4. The authors' response acknowledges that "tradeoff-breaking events cannot be regarded as sufficient conditions for an ETI", and the revised manuscript acknowledges (line 654) that "…a tradeoff-breaking event in and of itself cannot be used as a marker of an ETI." However:

– The Abstract (line 42) reads “tradeoff-breaking stands as a useful marker for ETIs,”.

– Line 537 that "…tradeoff-breaking observations are the mark of significant genetic innovation and, thus, can be used as a hallmark of ETIs,".

– Lines 417-418 that "…one genuine marker for an ETI is the capacity for a lineage to break away from such a tradeoff,".

– Lines 669-670 that "a general marker of ETIs to use in lieu of fitness decoupling is the emergence of rare tradeoff-breaking mutations."

– Lines 687-689 suggest that trade-off breaking is superior to fitness decoupling because "a fitness decoupling observation is not a reliable marker that an ETI is occurring". Since trade-off breaking is also not a reliable marker that an ETI is occurring, the point of this contrast is unclear.

5. I do not see how the conclusion that "…relative changes in fitness F and f2 do not explain the dynamics observed" follows from the rest of the paragraph (lines 407-417).

– For example, "the opposite directions of the dynamics of F and f2 are a consequence of optimal trait values on a tradeoff being different in different environments, not an inherent relationship between them" describes Michod's models neatly, with the dynamics of group fitness (F) and counterfactual fitness (f2) being a consequence of optimal trait values (investment in fecundity, investment in viability) being different in different environments (within the group versus outside the group). This seems to be an example, not a counterexample, of relative changes in fitness in F and f2 explaining the dynamics observed.

– In the snowflake yeast example, "f2 decreases…when whol*eLife* cycle fitness F increases" again seems to be an example of relative changes in fitness in F and f2 explaining the dynamics observed, but it is presented as a counterexample. Trade-offs and fitness decoupling in the counterfactual sense are not mutually exclusive explanations; the fact that trade-offs are involved, as in Michod's fitness decoupling models, does not mean fitness decoupling in the counterfactual sense is not occurring.

Recommendations for authors

These have a 1:1 correspondence with the items above

1. Adequately address Michod and colleagues' "counterfactual fitness" interpretation of their own model or give a compelling reason not to in the manuscript.

2. Make a compelling case for the lack of utility of the counterfactual fitness version of fitness decoupling in the manuscript or remove this claim from the manuscript.

3. Consistently describe what is and is not novel about the finding of incommensurability.

4. Consistently acknowledge that trade-off breaking is insufficient as a marker for an ETI.

5. Clarify, in the manuscript, how the dynamics of F and f2 fail to explain the dynamics described in lines 407-417.

*Reviewer #3 (Recommendations for the authors):*

I really enjoyed reading the revised manuscript, especially the newly-added Section 5. I think everything is much clearer now, and although parts and ideas have been discussed elsewhere, bringing them together in a coherent conceptual and theoretical framework is novel and useful. It certainly got me thinking in directions that I hadn't previously considered.

I have only one final comment. I appreciate the idea of separating the stage of collective formation from the subsequent ones and the authors' clarification that they focus on the latter (and hence that they choose not to focus on the competition with unicellular ancestors). But what if some collective property is needed precisely to overcome the competition with the unicellular ancestor? That's certainly how the experiments in Ulrich et al. 2018 came about: because in a model similar to the one in Tarnita et al. 2013 but published in the context of eusociality in Nowak et al. 2010, we predicted that a necessary condition for eusociality to evolve was for per capita fitness to get a drastic boost at very small, incipient group sizes, without any time for additional mutations. And in 2010 we had no clue what that could be. In 2018 we showed experimentally that spontaneous, rudimentary division of labor could do it.

In other words, one might sometimes be in a situation where Stage 2 is altogether skipped precisely because only collectives that snap right into some form of Stage 3 already during stage 1 can outcompete the ancestor. Put differently, Stage 1 = Stage 3.

Again, it seems to me that there are many ways to skin a cat and the authors are making it clear now that rare mutation doesn't have to be the only route to an ETI, but that some overarching framework based on traits and tradeoffs might offer a more productive way to look broadly at ETIs. I just wanted to alert them to the possibility that tradeoff breaking might not necessarily always occur after Stage 1 (and the associated competition with the ancestor) but sometimes might even occur during it. Unless I'm missing something?

---

## [Author Response]

Essential revisions:Fitness decoupling is an important topic that has created a lot of confusion in the field over the last decade and a half. Work clarifying this topic, and moving beyond looking at fitnesses, is welcome and important. The authors are well-poised to write this paper, bringing together real strength in the philosophy of biology, mathematical modeling, and experimental evolution of multicellularity. The modeling framework used here is useful, showing that cell and group-level fitnesses cannot be decoupled when they are both assessed over the same time scale. I agree with referee 3 that it is great to see the authors wrestling with real world data, and not relying only on simple mathematical models, because a key piece of the utility of this kind of theory will rely on people being to apply it to empirical systems. I also agree with the authors that traits, not just fitnesses, can be useful in determining when an ETI has occurred. Moving into traits, the authors propose to focus on trade-off breaking as an indicator that an ETI has occurred, which is creative and potentially quite useful, though further clarification will be required.The referees raised several important points that will need to be addressed in the revision.1) As Referee 1 points out, the idea that cell and group-level fitnesses cannot be formally decoupled was noted by Michod in his 2005, 2014, and 2020 papers. While I do not think that Michod has been entirely clear on this, his 2020 paper makes this point unambiguously. I think there's room for grace here with respect to what Michod thinks, and I'd like to avoid un-necessary strawman arguments. I don't think we should conflate Michod's views on this with Okasha's, and should give credit to Michod for accepting that cell and group-level fitness are inexorably correlated. But perhaps more importantly, the conclusion that in simple mathematical models cell and group-level fitnessess cannot be decoupled is not itself novel: Dr. Bourrat himself published this result as the central message of his 2015 paper on fitness decoupling (Bourrat, "Levels, time and fitness in evolutionary transitions in individuality". Philosophy and Theory in Biology). So I'd recommend modifying the language in the MS here around whether this is a novel finding (i.e., 2nd lead sentence of the discussion: "Our first main finding is the impossibility of decoupling between commensurable measures of fitness".).

Essential revision #1 concerns the novelty of the idea that commensurable measures of fitness cannot be decoupled and our treatment of previous publications in the domain. We do agree with the reviewers that this idea has already been developed in the literature and that a better description of our specific contribution is to add a new formal argument supporting this claim through mathematical modelling (something that was not done in Bourrat, 2015, for instance) and under more general conditions (i.e., we do not only address ‘simple models’ but all models that reach a stable collective size distribution).

Our intention is in no way to create strawman arguments. We agree that Michod and collaborators have clarified their position in several publications. In particular, we recognise that they (1) mention that particle and collective fitness must be correlated and (2) judiciously introduce counterfactual fitness to solve this problem. However, we argue that the export-of-fitness metaphor implies that the two quantities between which a fitness transfer occurs are commensurable. As a result, we think that the export-of-fitness model is incompatible with the use of counterfactual fitness (see our response to Reviewer #3’s public review for more details).

In light of this comment, we made the following changes to the manuscript:

We edited the wording in several places to distinguish the contributions of Michod and Okasha. We note that Okasha relies extensively on Michod and colleagues’ work to propose his three-stage model.In the glossary, we removed the mention of counterfactual fitness being a ‘recent development” in the export-of-fitness literature.In the introduction and Section 1, we clarified that our argument about fitness decoupling is to show *formally* its impossibility under reasonable assumptions, which is a novel contribution and not covered in Bourrat (2015).In the discussion, we replaced ‘Our first main finding is the impossibility of decoupling between commensurable measures of fitness’ with ‘Our first main finding is a new formal argument cementing the position holding that decoupling between commensurable measures of fitness is impossible’.

2) I think the main contribution of this paper is the shift from focusing on fitnesses to traits. Specifically, the authors advocate for the concept of trade-off breaking as an indication that an ETI has occurred. I like this idea, in part because it is far easier to measure traits than fitnesses in real organisms. However, the referees have raised some important issues with this idea that should be addressed during revision. As Referees 1 and 2 pointed out, organisms evolve trade-off breaking traits all the time without affecting biological individuality. I thought Referee 1's example of the zebra herd foraging trait was quite effective here, illustrating how collective-behaviors that break trade-offs may be not meaningfully affect the level of biological individuality.Most importantly, I'd like to see this idea developed a bit more to describe the conditions under which trade-off breaking is an indication of an ETI. Specifically, I'd like the authors to clarify: (1) Is every case of trade-off breaking a case of an ETI? (2) If not, are all ETIs characterized by trade-off breaking? Or (3) Is trade-off breaking one possible way of driving an ETI, but not a universal trait of ETIs? (4) If the latter is true, and trade-off breaking is not necessary or sufficient to distinguish an ETI, then what else is needed to distinguish an ETI? The authors have made a compelling case that the fitness decoupling literature has significant problems, but it is not yet clear that trade-off breaking is the solution.To clarify my thinking on this a bit more: trade-off breaking seems like a useful tool when there are clear trade-offs that can be ameliorated by multicellular adaptation (for example, by cellular differentiation- the cyanobacterial example is a good one). But what about a case of simple multicellular adaptation in the absence of pre-existing trade-offs? For example, consider a simple organism like a snowflake yeast that is obligately multicellular, reproducing by group fragmentation. Over generations, cells evolve novel traits that improve the fitness of cells living in multicellular groups, but these traits reduce the growth rate of cells. It's easy apply the counter-factual approach and see how the same traits increase the fitness of groups and the cells living in these groups, while decreasing the counterfactual fitness of solitary cells (they grow slower). To me, this is a straightforward use of the counterfactual fitness approach, in line with what Michod suggests in their 2020 paper, and is indicative of an ETI in progress. This isn't hypothetical, it's precisely what we see in snowflake yeast (see Bozdag et al., 2021 Nature Comms for some published data, but we have much more work done in this area that is unpublished). I don't think our model system is very unusual here, and I would think that this kind of simple group-level adaptation is common during the transition to multicellularity.Thus, the 3 referees and I agree that it is essential that the authors clarify whether trade-off breaking is a universal trait of ETIs, or is one potential diagnostic trait that an ETI has occurred. If it is not universal (which is what we suspect at this point, and claims of universality would require new, compelling arguments), then it would be useful to outline the conditions under which one might want to use a trait-based argument (i.e., trade-off breaking) vs. a fitness-based argument (i.e., counterfactual fitness). The referees and I generally thought that trade-off breaking could be a useful tool for determining that an ETI has occurred, even if it is not a diagnostic feature of all ETIs.

Essential Revision #2 concerns the generality of the tradeoff-breaking mechanism for the study of ETIs and asks us to detail our framework in the examples of the zebra herd and the snowflake yeast. We will start by answering the first set of concerns: the example of the zebra herd is addressed in our detailed answer to Reviewer #2 and we will come back to the snowflake yeast later.

As pointed out by the reviewers, we hope first and foremost that this manuscript will contribute to shifting the field from focusing on fitness to focusing on traits. We are motivated in this endeavour by our conviction that we, as a field, have reached the limits of what we can learn from fitness-based approaches to ETIs. One of the undeniable appeals of the export-of-fitness model is that it can be applied in the abstract without taking into account the likely very particular underlying constraints of the system. However, this constitutes a double-edged sword—when the assumptions of the export-of-fitness model are not met, the theory cannot account for the dynamics observed. This is so because, to expect a dynamic in which counterfactual fitness decreases and whole life cycle fitness increases, one must assume three things. First, the two quantities must depend upon the same traits; second, these two quantities are maximised for different trait values; third, the ancestral trait values (as the product of previous selection on the unicellular ancestor) correspond to the maximum of counterfactual fitness within the constraints of the ancestor. In contrast, if traits and tradeoffs are the centre of the theory, not only are all previous results about counterfactual fitness and ratcheting valid (albeit not explained using the metaphor of fitness transfer) but also tradeoff-breaking events—which occur when the three above assumptions do not hold—can be accommodated readily.

However, the reviewers are correct in their assessment that we did not discuss sufficiently the generality of the fitness tradeoff-breaking approach. First, if one takes the idea of tradeoff breaking in its broadest sense—that is, the emergence of rare mutations that overcome the usual constraints bearing upon a set of traits—we agree with the reviewers that tradeoff-breaking events cannot be regarded as sufficient conditions for an ETI. This is so because tradeoff breaking might occur without being accompanied by a change in the level of individuality. (Note that this is also the case with fitness-decoupling observations. Counterfactual fitness might decrease in a population of particles where particles interact with one another and must be surrounded by other particles to survive without this population producing higher-level individuals.) However, we suggest that it could tentatively be proposed as a necessary condition for the evolution of individuality. Several models of ETIs involve a first step of collective formation followed by collective level changes (e.g., Bourke, 2011; van Gestel & Tarnita, 2017; Rose & Hammerschmidt, 2021)—that is, a way to assemble lower-level entities in stable collectives—and secondary changes that cement the individuality of the new collectives. Tradeoff-breaking could be used as a practical way of assessing this second step, characterising what differs from collectives exhibiting low or little interaction (e.g., herds of deer, to take a famous example) and, thus, forming a necessary condition for a complete ETI.

The question of the snowflake yeast system is of particular interest to us. However, to treat the topic with the respect it deserves would require a detailed treatment that would go beyond the scope of this article. Nonetheless, we can make the following conjecture: the first snowflake mutation (ACE2) acts as the ‘collective formation‘ mutation (step one, similar to the WS mutation in the *Pseudomonas* system). From then, the observed change in cell shape (day 4 to 65 in the original experiments described in Ratcliff et al., 2012, 2013, 2014) fits well with the ‘optimisation on the tradeoff’, as we conceptualise it. As explored by Bozdag et al. (2021), the bigger clusters suffer from poorer oxygenation of the cells reducing the cell growth rate, which means that the increase in cluster size is accompanied by a reduction in within-collective particle growth rate (f_1_). A more delicate experimental question would be to establish whether the following recorded innovations (the changes in cluster shape in days 65–227 described in Ratcliff et al., 2013; the increased apoptosis described in Ratcliff et al., 2014; Jacobsen et al., 2018; or even the snowflakes aggregates described in Gulli et al., 2019) constitute a proper ‘tradeoff breaking‘ as we describe it in the article. One way to explore this would be to combine the existing data on cell and cluster shape (Ratcliff et al., 2012, 2013, 2014), physical models of cluster volume (Libby et al., 2014; Jacobsen et al., 2018), simple sedimentation models (as used in Solari et al., 2015), and within-flake cell growth models (Bozdag et al., 2021) and give the snowflake system the same transversal treatment that we give the *Pseudomonas* system in this article.

In light of these comments, we made the following changes to the manuscript:

We have added a new Section 5 that (1) makes clearer that tradeoff breaking is the second (or third) step in an adaptive scenario that involves ‘collective formation’, ‘optimisation on the tradeoff’ and ‘tradeoff breaking’. To make this effective, we have added a new figure (Figure 8), (2) discussed the generality of the tradeoff approach, and (3) contextualised our approach within the broader picture of ETIs, linking it to the existing framework.We now include the snowflake yeast system in our examples (in Section 5).

3) Referee 3 points out that the model developed in this paper is not itself novel, being similar in structure and analysis to her prior work. Please cite/discuss this prior work appropriately.

Essential Revision #3 concerns the novelty of the modelling approach. We do agree that we are not the first to have proposed to model proto-collectives using size classes and transition between those size classes (growth, shrinkage, and propagule production). The model can be traced back to an extension of Leslie matrices (Leslie, 1945) and their generalisation as class-based Matrix population models (Caswell 1989). The literature on this topic already makes use of Perron-Frobenius theory to make asymptotic predictions.

Tarnita et al. (2013) present a valuable treatment of the problem using ordinary differential equations and focus on competition with the single-cell counterpart, which we ignored for the purpose of this article. Pichugin et al. (2017) uses a similar dominant eigenvalue method (as we did) to treat the problem of complex fragmentation modes (whereas we only allow single-cell propagules shedding).

In light of these comments, we have better situated our modelling approach with respect to the existing literature by making the following changes:

We added the following references to the paragraph introducing the model (Section 3): Leslie (1945), Caswell (1989), Tarnita (2013), and Pichugin (2017).We discussed in further detail the fact that we ignore competition with the unicellular ancestor as studied in Tarnita et al. (2013) (Section 5).

Reviewer #2 (Recommendations for the authors):Figure 2 would be more useful if it included a comparison of fitnesses, for example between different genotypes (in other words, if it compared particle- and collective-level fitnesses of two genotypes and showed that they are identical if measured over the same time span).

We have decided not to change Figure 2, as it is introduced in Section 1 where we have not yet presented the idea that fitnesses measured at the particle and collective level are necessarily equal. We believe this could potentially confuse the reader.

Reviewer #3 (Recommendations for the authors):Let me start by saying the I am sympathetic to the authors' position on fitness decoupling, which I've always found to be so contrived as to be either non-sensical or meaningless, depending on how one defines fitness. I also quite like the simple proposals of sections 3 and 4 and appreciate that the authors engaged both with empirical evidence and with ways to see existing theoretical models through the lens of the tradeoff-breaking framework. This being said I have a few concerns that I would like addressed:1. the writing is fairly dense and convoluted: it took me several hours to go through the manuscript, and I'm still unsure that I'm fully grasping all details. If this is going to be engaged with by a broad audience, and especially a non-specialist one, the authors need to make more of an effort to clean up, streamline, and provide intuition. For instance,– I can't make sense of Figure 4 I don't know which ones of the dots are particles; I don't know which is the focal particle whose fitness we are tracing, I don't understand what the vertical dotted lines are, and I don't know why there are multiple horizontal lines, seemingly associated with both blue and orange dots. This lack of clarity around exactly how to measure f2 and F propagate, making it sort of hard to rigorously understand what the authors meant f2 to be in various of the examples (though, in principle, if I don't worry too much about understanding Figure 4, I can follow what f2 in principle might be in each example, as needed).

We have extensively reworked Figure 4 to make this clearer.

– I found it tedious to go through all the math derivations and references to theorems in the main text, with, often, little to nothing by way of biological intuition that would help ground some of those results. When some biological intuition is provided, it often left me wanting some more explanation. For instance, in discussing the example of section 3.3, the authors state: "Note importantly that the opposite directions of the dynamics of F and f2 is a consequence not a cause of the underlying tradeoff mechanism that drives the evolutionary dynamics." It would be great if the authors could explain why this is; and, in general, this whole example could use a bit of biology on the very bare theoretical bones. For example, it seems that all collectives of size i >= 2 have the same survival probability and the same probability to grow to the next class size; if the authors can't contextualize this with a bit more biology, at the very least it would be nice to have a sentence that simply states this assumption in words. A non-mathematical reader might have a hard time inferring that from the fact that the expressions of p_i and g_i don't depend on i.

We have now provided further explanation and a biological example to address this point.

– Although I understand the tradeoff in Black et al., I'm still not entirely sure I understand the tradeoff in the model in Box 2: is there supposed to be a tradeoff linking β and q? It seems that β is a growth rate whose increase would be good for both germ cells and soma cells. In both Boxes 2 and 3 it would be helpful to have it more clearly spelled out which modeling aspects are new to this paper and which ones are taken from the original papers, which were themselves modeling papers.

In the model from Black et. al, the trait β is more akin to the underlying resource allocation trait that can be seen in other tradeoffs models, such as the trait θ from Section 3 of the main text. It corresponds to the linear abscissa on a 2D tradeoff line. The tradeoff exists in the sense that a collective cannot at the same time have a high initial growth rate β and reach a high collective size at the moment of dispersal, *g(T).* This tradeoff is broken by mutations on *q* because somatic cells improve dispersal (if ρ>0) while reducing less the growth rate than mutations on β .

We made the following edits to the boxes:

We explicitly mentioned the parallel between β . (from Black et al.) and θ (in the main text) to clarify the tradeoff involved in this example.We made more precise references to the original equations,We added a sentence: ‘we now recast the model within our framework’ to delineate the point where we start to make an original contribution.

– Simultaneously, some things that are simple to state intuitively, are presented in a somewhat convoluted and tedious way: e.g., the 1st paragraph of Box 1 and associated Figure B1a make overly complicated the story that any reader would appreciate with a simple example: a bacterium that doubles every hour will make more of itself in one year than a mouse that "doubles" every year (or interest compounding or a suite of other simple, intuitive examples). I also think that some such intuitive discussions should come earlier, as the authors set up the discussion of fitness in the main text (prior to Figure 4).

We have now streamlined the first section of Box 1 and provided a biological intuition using an example involving bacteria.

– In general, I'm not convinced that some of the mathematical detail needs to be in the main text as opposed to a detailed methods section.

We have decided to keep those details in the main text. However, we provided some signposting at the beginning of Section 2 so that the reader willing to skip the mathematical details can easily do so.

2. The model in Section 2 is not new. Tarnita et al. 2013 J. Theor Biol (Evolutionary construction by staying together and coming together) have used the same model to explore the origin of multicellularity. Even though that paper looks at competition of such collectives against the solitary ancestor, the ideas, setup and analyses are nearly identical to what the authors do in section 2.

Thanks for bringing this to our attention. See our response to Essential Revision #3.

3. This brings me to my final and broadest points:– First, if I understand correctly, in this manuscript Bourrat et al. do not seem to consider competition against an extant unicellular ancestor anymore. But, it seems to me that one of the greatest challenges would've been the persistence of the multicellular lifestyle in the earliest stages, before rare mutations can break tradeoffs and whatnot. It would be useful to at least have some discussion of this early on, so that I can situate where exactly the modeled worlds are in the origin of multicellularity and transition to individuality (i.e. what is assumed to have happened already, vs what is the new challenge to be explained).

Our main goal is to define a unified framework where different fitness measures that have been used experimentally (e.g., counterfactual particle fitness, within-collective particle fitness, whole–life cycle particle fitness) can be described and their relationship studied in a mechanistic way. In particular, we want to be able to avoid making reference to the export-of-fitness metaphor, on which we share the reviewer’s opinion.

The paper assumes that some early ‘collective formation’ (WS, ACE2, etc.) mutation, or environmental change has already happened and focuses on the early evolutionary dynamics of the proto-multicellular organisms. It also assumes that nascent multicellularity will be constrained by tradeoff inherited from the unicellular ancestors, and that underlying constraints explain in part what has been described as fitness decoupling. However, we advocate that this pattern does not hold as soon as tradeoff-breaking mutations arise (something that presents a challenge for the export-of-fitness view).

In light of this comment, we have added a sentence at the beginning of Section 2 (where we present our model) to state that we do not study the potential competition between particles and protocollectives, as done in Tarnita et al. (2013). We return to this point in Section 5, which also better situates our work with respect to the different phases of an ETI.

– Second, how should we think of free riders in this framework? They don't seem to be explicitly included in any of the modeling.

Free-riders, or cheaters, are not the focus of this model, but could be addressed if necessary using an additional layer of game-theory modelling. We now explicitly mention this at the beginning of Section 2.

– Third, much as in the original Michod thinking, it seems to me that the current framework also sort of assumes that the collective is a sum of what the component cells do and does not consider spontaneous emergent properties that simultaneously confer benefits of group living on the group and also on the parts, such as some degree of division of labor [that harnesses the repertoire of the unicellular ancestor and arises e.g., due to emergent information gradients in the earliest collectives]. Such spontaneous benefits could in themselves set the stage for an ETI without requiring fundamental reversals of tradeoffs (even though these of course could follow later). Again, I'm not suggesting that the current framework should be explicitly modified to account for such dynamics, but it would be informative to have at least some discussion of whether the authors believe that tradeoff-breaking by rare mutations is the only way to get an ETI or whether it's one possible thing to look for empirically, but not necessarily the only route.

We are interested in the same topic as the reviewer here: how the complexity of the repertoire of the unicellular ancestor affects the early collectives. We think that focusing on traits (and tradeoffs) instead of fitness is a necessary step in addressing this question. We do recognise that phenotypic plasticity could play a driving role in this phenomenon, leading to apparent tradeoff breaking without requiring mutations. We have included this possibility in a new paragraph within section 5 (beginning with ‘Yet another possibility not studied in this article is…’).

To contextualize this point further, it might be helpful to look at Ulrich et al. 2018 Nature (Fitness benefits and emergent division of labour at the onset of group living) and Brahma et al. 2017 (Emergence of cooperation and division of labor in the primitively eusocial wasp Ropalidia marginata). Both of these look at the evolution of eusociality rather than multicellularity, but the essence of the ETI can be captured by the same model (see, Tarnita et al. 2013). I'm not aware of similar empirical work in multicellularity, though see Gavrilets 2010 PLoS Comp Biol (Rapid Transition towards the Division of Labor via Evolution of Developmental Plasticity) for theoretical work.

Ulrich et al. (2018) and Tarnita et al. (2013) are now cited. We also have cited Ispolatov et al. (2012), which is relevant in this context.

*[Editors' note: further revisions were suggested prior to acceptance, as described below.]*
The three referees and I have read your revised manuscript, and really appreciate some of the new additions. In particular, there is a lot of enthusiasm for the new Section 5. Everyone found the paper to be thought-provoking and thought that this paper will make an important contribution to understanding ETIs if suitably revised.The main thing we would like to suggest is a more inclusive tone for the paper. I do not think it takes away any of the novelty of the results, and indeed, can strengthen the scholarship of the work. It is critical that you revise the introduction to more fairly reflect Michod et al.'s stated meaning (not the authors' inferred meaning) about the concepts of export of fitness, fitness decoupling, and the counterfactual account. Second, the referees and I do not see why the counterfactual account is not a valid indicator that an ETI has occurred. This does not undercut the utility of your work, showing that trade-off breaking mutations can be an indication of an ETI. But all four of us are convinced that both approaches have merit.Referee 2 has left a detailed review, highlighting five major points. The referees and I agree that it is essential to address all five points with changes to the MS, not arguments for why revision is not necessary. Please respond to their review thoroughly. I have carried forward their five summary points and provided a bit of additional context on some below, reflecting our collective post-review discussion.

Thank you for your detailed report. Our response to the comments is written in purple below. We would like to note at the outset, however, that we do not see the counterfactual account, in and of itself, as a problem. Our disagreement with the literature is limited to the concepts of export, decoupling, and transfer of fitness: we believe that the counterfactual account, while fruitful, does not satisfactorily address their limits. This response letter and the changes we have made to the manuscript aim to make this point unambiguously clear.

To address the comments from Reviewer 3 and the Reviewing Editor, we have pasted each of their five points together and responded to both in a single response (in purple below). Please find enclosed a new version of the manuscript, edited in light of your comments.

1. Adequately address Michod and colleagues' "counterfactual fitness" interpretation of their own model or give a compelling reason not to in the manuscript.Despite pointing this out in the last revision, the paper still does not treat Michod et al.'s position on fitness decoupling and counterfactual fitness fairly. The paper is framed in such a way that it critiques a straw-man version of Michod et al's position on these topics (privileging the authors' interpretation of what Michod et al. have said, or what the metaphor of "fitness decoupling" implies), which stands in contrast to what Michod and colleagues have explicitly said they mean by these terms in their more recent papers.

Thank you for these comments, which have led us to revise the way the paper is framed. It was not at all our desire to represent ideas expressed by others unfairly. Likewise, it was not our intent to attack a strawman version of the export-of-fitness account. Rather, we aim to address this concept seriously and follow through with what it implies. We sincerely regret not having been able to convince the referee of our good faith.

We must stress that, in principle, we have no disagreement with the counterfactual approach proposed by Shelton and Michod (2014), which was already presented more implicitly in some earlier work by Michod and colleagues. However, our main contention is that it is not compatible with the ‘export-of-fitness’ account. This was stated in Line 715 of the second submission: ‘The method described by Shelton and Michod (2020), once separated from the export-of-fitness model, would prove useful here’. Here is our rationale why this is so.

The notion of counterfactual fitness stands on its own without having to be tied to fitness decoupling/transfer/export. This point can be vindicated from the work of the proponents of the counterfactual approach. In their 2014 article, Shelton and Michod mention neither fitness transfer nor fitness decoupling, thus leaving unresolved the connection between counterfactual fitness and notions of fitness decoupling, transfer or export. The same applies to the follow-up article by Leslie et al. (2017). The connection between the ‘fitness decoupling’ account and ‘counterfactual fitness’ was only later made explicit by Shelton and Michod (2020), where the authors state (in the abstract) that ‘during an ETI, fitnesses at the two levels, group and individual, become decoupled, in the sense that fitness in a group may be quite high, even as counterfactual fitness goes to zero.”

Thus, it does appear that Michod and colleagues have proposed two versions of the export-of-fitness account. To our knowledge, they have not explicitly stated that one replaced the other (however, Reviewer 3 seems to disagree on this point). Regardless of whether they have been explicit, in the interest of presenting a complete picture of the problem, we have decided to present and discuss both versions.

The earlier account was not explicit about the counterfactual nature of fitness and can be (and has been) interpreted as relating two ‘actual‘ fitnesses. We oppose this interpretation of fitness transfer/export/decoupling because actual cell and collective fitness cannot *actually* be decoupled, as was also appreciated by Shelton and Michod (2014). To this end, we prove mathematically that these fitnesses are equal for a wide range of models (we return to this point in the response to Essential Revision #3).

The second account explicitly refers to ‘counterfactual‘ fitness. We oppose this interpretation of fitness transfer/export/decoupling because there is no meaningful ‘transfer‘ possible between counterfactual and actual quantities. We justify this point via two arguments. First, by definition, a counterfactual fitness does *not* exist—thus, there is nothing material that can be transferred from a counterfactual to an actual quantity. Second, since fitness is not a conserved quantity in the model, there is no total fitness that could be distributed and transferred between potential and actual fitnesses, as is the case for energy in physics. Nevertheless, we recognise that counterfactual quantities are valuable for other purposes.

We think that making these points is not superfluous because the metaphor of fitness transfer is difficult to apply to experimental systems and has been used in inconsistent ways. In some of his most recent writings, Michod seems to agree that the straightforward interpretation of the term ‘transfer’ raises difficulties: ‘speaking of fitness “transfer” may suggest that fitness is a conserved quantity in the model which it is not’ (Michod, 2022, p44, our emphasis). As an example, when discussing fitness transfer, Shelton and Michod (2020) discuss parameters in Hamilton’s rule in the following way: ‘the evolution of altruism transfers fitness from the level of the cells to the level of the group’. They reason that this is so because the cost of altruism (C ‘units of fitness’ in Hamilton’s rule) is ‘subtracted from the cell level’ while its benefits (B ‘units of fitness’) is ‘added to members of the group’ (Shelton & Michod, 2020, p7). However, we would like to point out that the parameters C and B do not correspond to counterfactual cell and group fitness: they both appear in the expression of collective fitness (i.e., the cost C is paid in both the context of the collective and outside of it) making them de facto actual rather than counterfactual quantities.

That being said, we do agree with the reviewer that our original manuscript failed to make these points clear and could be interpreted, at times, as a rejection of counterfactual fitness in its entirety. To clarify our position, we have now reworked the Introduction to:

Present the idea of counterfactual fitness that we credit to Shelton and Michod (with mention of Michod’s earlier work, in which the idea is more implicit).Link counterfactual fitness, following Michod and Shelton (2020), to fitness decoupling/transfer.Recognise the general value of counterfactual fitness, but highlight the difficulty when it is used to explain fitness transfer/decoupling.

Further, in Section 1, we now provide several paragraphs presenting our rationale regarding why fitness decoupling/transfer, even if understood from a counterfactual perspective, is not adequate.

In the Discussion, we have added a section on the usefulness of counterfactual particle fitness in the context of ETIs and how it can be integrated into the tradeoff-breaking framework.

2. Make a compelling case for the lack of utility of the counterfactual fitness version of fitness decoupling in the manuscript or remove this claim from the manuscript.Given that you are using a Darwinian Individuality approach to define what is an individual (clearly defined in the glossary, thank you!), I do not see how one can dismiss the utility of the counterfactual fitness approach. Empirical measures consistent with the counterfactual fitness account can provide insight into whether groups have been acting as Darwinian Individuals. For example, consider a lineage evolving along a pre-existing trade-off between traits that improve the fitness of groups and those that improve the fitness of cells, like those identified in Figure 8's depiction of 'optimization on the trade-off'. Using the example given in this section, consider the mutations that make snowflake yeast cells elongate, which increase group size (and thus fitness) but decrease cell growth rates (and thus fitness). This would be detected in the counterfactual fitness account and would rightly be interpreted as a trait that evolved due to selection acting among groups, not among individual cells, demonstrating that groups are Darwinian Individuals. Trade-offs have not been broken, but nonetheless, groups replicate, have heritable variation in group-level traits, and selection acts on those traits, resulting in adaptations that would not have occurred among unicellular populations. I think we can all agree that in this case, measurements consistent with the counterfactual approach would indicate that groups are a Darwinian Individual according to PGS's criteria, right? On a related note, I couldn't figure out why the MS embraced the 'Ratcheting' model but rejected the counterfactual account since ratcheting is based on the same trade-offs that would be detected by the counterfactual fitness account.However, I think your paper raises very valid limitations on the scope of counterfactual fitness- not all multicellular adaptations will necessarily create a trade-off between fitness in groups and fitness in isolated cells, reverting lineages to unicellularity may often prove impossible, and measuring fitness can be difficult, etc. From my perspective, both the counterfactual fitness account and the trade-off breaking account may be useful indicators that an ETI (following the Darwinian Individuality criteria) has occurred. And importantly, I do not think either is necessary or sufficient to show an ETI has occurred without additional information about the traits themselves. They both require context to be informative.Since neither is necessary for an ETI, a lineage may evolve along either route (or both simultaneously, as they are not mutually exclusive). I could imagine cases where it is much easier to identify trade-off breaking mutations and others where it is easier to identify counterfactual fitness. Having both tools available is a net positive for our field, and I don't see the need to limit ourselves only to tradeoff-breaking mutations. All the more so if trade-off breaking mutations are not necessary or sufficient to show an ETI, making the decision to only consider them somewhat arbitrary.

We wish to state again that our intention is not to dismiss the counterfactual approach. Instead, we are raising concerns regarding its interpretation in the context of the export-of-fitness (fitness transfer and decoupling) account. As mentioned in our response to the previous point, we find the idea of fitness decoupling/export/transfer inadequate in both its actual and counterfactual versions. Thus, with respect to counterfactual fitness, our main contention is with the *interpretation* of fitness decoupling/transfer from the counterfactual, not that the idea of counterfactual fitness can be relevant for explanation in ETIs. Our framework is compatible with both fitness decoupling and tradeoff-breaking observations and provides them a unified interpretation in terms of a tradeoff. We have reworked our manuscript to avoid suggesting that the second kind of observation should replace the first.

To ensure that our position does not come across as against the idea of counterfactual fitness, in addition to the point mentioned in the previous response, we:

Have made the point unambiguously that we find the notion of counterfactual fitness useful (throughout the manuscript).We changed the title of Section 2.4 and reworked the paragraph.We have also reworked the snowflake yeast example in Section 3.3, which clarifies the link between actual and counterfactual fitness.

3. Consistently describe what is and is not novel about the finding of incommensurability.

This is described clearly in the original review and I have no additional context to add.

Our work gives a general mathematical proof of the incommensurability of individual and collective fitness. It vindicates the point made previously by Shelton and Michod (2014) (‘cells and colony level fitness […] are mathematically equivalent ways of bookkeeping’(p. 457)), Bourrat (2015b) (‘during an ETI, if the fitness of the particles seems incommensurable with the fitness of the collective, it is most probably due to the fact that, during the last stage, those fitnesses are not measured over the same period of time anymore, and the interactions between particles become so complex that tracking back their fitness over longer periods of time than one or two generations appears in practice impossible’.) and Black et al. 2020 (‘within our model, this is readily explained: fitness of a cell is measured over the short timescale while patch fitness is measured over the long timescale. This captures precisely—and explains mechanistically—the notion of ‘fitness decoupling’ thought to occur during the earliest stages of the evolution of multicellular life but which has often been difficult to intuit’).

Our results are more general than the ones previously published because they do not rely on any a priori assumption regarding the relationship between collective and individual fitness, viability, and fecundity (linearity, average, tradeoff…), or on writing the model in a MLS1 or MLS2 framework. We show that in *any* population growth model where collective sizes reach a stable distribution, the long-term growth rates of particles and collectives are equal.

To address the reviewer’s points, we have made the following changes to the manuscript:

In the abstract, we have replaced the passage mentioned with ‘Extending and unifying results from the literature, we show that fitness of particles and collectives can never decouple because calculations of particle and collective fitness performed over appropriate and equivalent time intervals are necessarily the same provided the population reach a stable collective-size distribution’.”Lines 95–97 have been deleted.We have rewritten lines 109–110. We added that our work is “’generalising and unifying results from the literature” and provided references to acknowledge previous work.Lines 190–191 have been deleted.In Section 2.4, we have now explicitly stated the novelty of the generality of this result: ‘while these points have been made in the literature (see Section 1) our mathematical proof applies to a larger class of population dynamics models—it is not tied to a MLS1/MLS2 framework and does not require any assumptions regarding the relationship between particle and collective life history traits, such as the usual assumption that collective viability is a linear function (since the γ or μ,δ, and β are free). It only requires that the population reaches a stable size distribution; only if the collectives are able to grow (or shrink) indefinitely, which is not a realistic assumption for ETIs, could genuine fitness decoupling be observed (see Bourrat, 2021b, Chapter 5 for details).’Recent works by Shelton and Michod are now cited throughout the manuscript, as described in other essential revisions.

4. Consistently acknowledge that trade-off breaking is insufficient as a marker for an ETI.As acknowledged in the prior review, trade-off breaking is not sufficient to determine that an ETI has occurred. It is possible to have a trade-off-breaking mutation occur within a single level. The referees and I agree that the MS should be consistent about this limitation and avoid saying the existence of that trade-off breaking mutations mean an ETI has occurred. To this end, it would be very helpful if the authors could identify what additional information one would need to know in order to conclude that a trade-off breaking trait indicates that an ETI has occurred (i.e., it's pretty clear if the adaptation derives from cellular differentiation in a way that overcomes cell-level trade-offs, like C and N fixation in cyanobacteria, but what about other kinds of adaptations? How would one infer that adaptations generating novel multicellular morphology are breaking trade-offs? What if the adaptations in question do not stem from the limitations of the unicellular ancestor [i.e., those that are limited by trade-offs], but rather reflect the novel ecology and behavior of the multicellular organism?).

Thank you for pointing out that we were not sufficiently clear about the role of tradeoff breakings in ETIs.

We agree that tradeoff breaking can occur within a single level. This was stated in the sentence before the one quoted by the reviewer: ‘“constraints breaking” are already considered key events in the evolution of body plans and are expected to be a widespread mechanism for the emergence of novelties’ (L 652). However, we suggest that the observation of tradeoff breaking in the context of an ETI (i.e., on traits of interest) is a relevant event that can be used as a marker of an ETI even if it is not sufficient.

Our model aims to be compatible with both fitness decoupling and tradeoff-breaking observations. We have reworked our manuscript to avoid suggesting that the second kind of observation should replace the first; rather, it should come at a later stage of the transition (phase 3 instead of phase 2 in Figure 8).

Regarding the operationalisation of the concept, please note that, in our framework, the tradeoff-breaking mutation is an abstract phenotypic change that could, in principle, be due to ecology, behaviour, or plasticity, in addition to genetic mutations. Developing a general method to select relevant traits and detect tradeoff breaking in any transition in individuality moves beyond the scope of the present manuscript, but is a promising avenue for future research.

To clarify these points and address the comments from the Reviewer and Reviewer Editor, we have:

Changed the sentence in the Abstract (line 42) from “tradeoff-breaking stands as a useful marker for ETIs” to “Thus, when observed in the context of ETIs, tradeoff-breaking events stand as a useful marker for these transitions.”Changed the sentence on line 537 from “… tradeoff-breaking observations are the mark of significant genetic innovation and, thus, can be used as a hallmark of ETIs,” to “as we advocate that such tradeoff-breaking observations in this context are the mark of significant innovation and, thus, can be used to detect ETIs.”Changed the sentence in lines 417–418 from “…one genuine marker for an ETI is the capacity for a lineage to break away from such a tradeoff," to “the capacity for a lineage to break away from such a tradeoff could be used to detect an ETI.”Lines 669–670 have been rewritten and now read as: “Finally, we propose that rare tradeoff-breaking events are a crucial part of ETIs and could be used experimentally for their detection.”Lines 687–689 now read: “Our second main result is a general adaptive scenario of ETIs that is compatible with both experimental fitness-decoupling observations (artifactual or counterfactual) and tradeoff-breaking deviations.”Given more context throughout the manuscript and following the response to the previous comments regarding why tradeoff breaking can be a marker for an ETI.In Section 5, we have:Italicised “in and of itself” in the sentence: “Thus, a tradeoff-breaking event in and of itself cannot be used as a marker of an ETI.”Contextualised our claims in the following: “However, considering a trait in the context of an ETI, where collectives are formed and maintained (phases 1 and 2) and establishing how those collectives change and potentially overcome the constraints that historically bear on particles (phase 3) offers a good empirical handle to study ETIs. Conversely, the lack of any tradeoff-breaking observation in such an empirical system could be used as an indication that the transition is still in its early stages.”

To answer specifically regarding the other information necessary for an ETI, we would need to have clearly identified discrete higher-level entities (or entities in the process of discretisation), what Clarke (2013) calls “mechanisms of demarcation.” We now mention this in Section 5.Finally, to clarify the scope of the tradeoff-breaking model, we have included a sentence at the beginning of Section 3 stating that we consider mutations to be abstract changes in phenotype that could potentially be induced by environmental changes (on top of genetic mutation and phenotypic plasticity).

5. Clarify, in the manuscript, how the dynamics of F and f2 fail to explain the dynamics described in lines 407-417.This is described clearly in the original review and I have no additional context to add.

There seems to be disagreement between us and Reviewer 3 regarding what constitutes an explanation of dynamics. We do not think that the relative dynamics of *F* and *f2* explain the evolutionary dynamics of the system because there is no actual causal link between the dynamic of the counterfactual fitness *f2* and the actual evolutionary dynamics. The value of *f2* is entirely irrelevant to the dynamics of trait values. Starting from a given initial condition (i.e., initial trait value), the trait would have the same dynamics whether *f2* decreases, increases, or even oscillates. The only quantity that matters and explains the trait changes is the gradient of F.

Now, measuring the counterfactual fitness *f2* might be of interest, but not because its dynamics relative to *F* explain anything about the dynamics of the trait value theta (e.g., through some putative transfer mechanism)—rather, because it gives us a hint that there might be a tradeoff here between traits that contribute to collective fitness or would contribute to counterfactual cell fitness.

Once again, our point is that the counterfactual fitness approach is valuable, but only when taken for what it is: a counterfactual, and not what it is not (and what it cannot be)—that is, something to be transferred to F. Using it in the context of export-of-fitness is inconsistent, as we have argued in the previous points.

We have edited line 407–417 to reflect this (this also partly addressed Essential Revision #2). We made a reference to this paragraph in Section 2.4 when we discuss the general usefulness of fitness-decoupling observations.

Reviewer #2 (Recommendations for the authors):Recommendations for authorsThese have a 1:1 correspondence with the items above1. Adequately address Michod and colleagues' "counterfactual fitness" interpretation of their own model or give a compelling reason not to in the manuscript.

This is now done in the Introduction, Section 1, and the Discussion.

2. Make a compelling case for the lack of utility of the counterfactual fitness version of fitness decoupling in the manuscript or remove this claim from the manuscript.

We have now clarified the link between counterfactual fitness and fitness decoupling/transfer, and why counterfactual fitness does not provide a satisfactory account of fitness decoupling. However, we have stressed that the concept of counterfactual fitness, in and of itself, can be useful in the context of ETIs.

3. Consistently describe what is and is not novel about the finding of incommensurability.

We have now clarified, from the abstract onwards, what is novel about our result.

4. Consistently acknowledge that trade-off breaking is insufficient as a marker for an ETI.

We have now done so.

5. Clarify, in the manuscript, how the dynamics of F and f2 fail to explain the dynamics described in lines 407-417.

These lines have been rewritten.

Reviewer #3 (Recommendations for the authors):I really enjoyed reading the revised manuscript, especially the newly-added Section 5. I think everything is much clearer now, and although parts and ideas have been discussed elsewhere, bringing them together in a coherent conceptual and theoretical framework is novel and useful. It certainly got me thinking in directions that I hadn't previously considered.I have only one final comment. I appreciate the idea of separating the stage of collective formation from the subsequent ones and the authors' clarification that they focus on the latter (and hence that they choose not to focus on the competition with unicellular ancestors). But what if some collective property is needed precisely to overcome the competition with the unicellular ancestor? That's certainly how the experiments in Ulrich et al. 2018 came about: because in a model similar to the one in Tarnita et al. 2013 but published in the context of eusociality in Nowak et al. 2010, we predicted that a necessary condition for eusociality to evolve was for per capita fitness to get a drastic boost at very small, incipient group sizes, without any time for additional mutations. And in 2010 we had no clue what that could be. In 2018 we showed experimentally that spontaneous, rudimentary division of labor could do it.In other words, one might sometimes be in a situation where Stage 2 is altogether skipped precisely because only collectives that snap right into some form of Stage 3 already during stage 1 can outcompete the ancestor. Put differently, Stage 1 = Stage 3.Again, it seems to me that there are many ways to skin a cat and the authors are making it clear now that rare mutation doesn't have to be the only route to an ETI, but that some overarching framework based on traits and tradeoffs might offer a more productive way to look broadly at ETIs. I just wanted to alert them to the possibility that tradeoff breaking might not necessarily always occur after Stage 1 (and the associated competition with the ancestor) but sometimes might even occur during it. Unless I'm missing something?

Thank you for your feedback—we are glad that our revisions added clarity. We agree that Stage 2 can be skipped. This is included in Figure 8 as Route 2 and also in the section on the importance of phenotypic plasticity for a primordial form of division of labour. We have modified the relevant part of Section 5 to clarify that Stage 3 can immediately follow Stage 1.

It must be noted that since the focus of this article has been predominantly on fitness-decoupling observations, we have explained Route 1 in more detail. Nevertheless, Route 2 is very interesting and might, in fact, be widespread in nature.